# DyG2Vec: Efficient Representation Learning for Dynamic Graphs

**Mohammad Ali Alomrani**[*]                                   *mohammad.ali.alomrani@huawei.com*
*Huawei Noah's Ark Lab*

**Mahdi Biparva**[*]                                           *mahdi.biparva@huawei.com*
*Huawei Noah's Ark Lab*

**Yingxue Zhang**                                             *yingxue.zhang@huawei.com*
*Huawei Noah's Ark Lab*

**Mark Coates**                                               *coates@ece.mcgill.ca*
*McGill University*

**Reviewed on OpenReview:** *https://openreview.net/forum?id=YRKS2J0x36*

## Abstract

Temporal graph neural networks have shown promising results in learning inductive representations by automatically extracting temporal patterns. However, previous works often rely on complex memory modules or inefficient random walk methods to construct temporal representations. To address these limitations, we present an efficient yet effective attention-based encoder that leverages temporal edge encodings and window-based subgraph sampling to generate task-agnostic embeddings. Moreover, we propose a joint-embedding architecture using non-contrastive SSL to learn rich temporal embeddings without labels. Experimental results on 7 benchmark datasets indicate that on average, our model outperforms SoTA baselines on the future link prediction task by 4.23% for the transductive setting and 3.30% for the inductive setting while only requiring 5-10x less training/inference time. Lastly, different aspects of the proposed framework are investigated through experimental analysis and ablation studies. The code is publicly available at https://github.com/huawei-noah/noah-research/tree/master/graph_atlas.

## 1 Introduction

Continuous-time dynamic graphs (Kazemi et al., 2020) are graphs in which each edge has a continuous timestamp and can be naturally found in many real-world applications such as social networks and finance. Recently, dynamic graph encoders (Rossi et al., 2020; Wang et al., 2021b; Jin et al., 2022; Luo & Li, 2022) have emerged as promising representation learning approaches that are able to extract temporal patterns from an ever-evolving dynamic graph in order to make accurate future predictions. However, such models have several shortcomings. First, they heavily rely on chronological training and/or complex memory modules to construct predictions (Kumar et al., 2019; Xu et al., 2020; Rossi et al., 2020; Wang et al., 2021b). Consequently, encoding any dynamic graph requires sequentially iterating through all edges, which is intractable for large graphs due to the high computational overhead. Second, the encoding modules either use inefficient message-passing procedures (Xu et al., 2020) that enforce temporal causality, or expensive random walk-based algorithms (Wang et al., 2021b; Jin et al., 2022) with heuristic feature encoding strategies that are engineered for edge-level tasks only. Finally, as opposed to other temporal domains (Tong et al., 2022; Eldele et al., 2021), most works on dynamic graphs have focused on pushing downstream task performance rather than learning general pre-trained models.

---

[*]equal contribution

Self-Supervised Representation Learning (SSL) has shown promise in achieving competitive performance for different data modalities on multiple predictive tasks (Liu et al., 2021). Given a large corpus of unlabelled data, SSL postulates that unsupervised pre-training is sufficient to learn robust representations that are predictive for downstream tasks with minimal fine-tuning. Contrastive SSL methods, despite their early success, rely heavily on negative samples, extensive data augmentation, and large batch sizes (Jing et al., 2022; Garrido et al., 2023). Non-contrastive methods address these shortcomings, incorporating information theoretic principles through architectural innovations or regularization methods (Balestriero & LeCun, 2022). The success of such SSL methods on sequential data (Tong et al., 2022; Eldele et al., 2021; Patrick et al., 2021) suggests that one can learn rich temporal node embeddings from dynamic graphs without direct supervision. While there are some recent attempts at using SSL for dynamic graphs such as DDGCL (Tian et al., 2021) and DySubC (Jiang et al., 2021), they tend to require high memory and computation due to negative sampling and focus more on pushing downstream performance rather than learning rich general representations.

In this work, we propose DyG2Vec, a novel efficient encoder-decoder model for continuous-time dynamic graphs that benefits from a window-based architecture that acts as a regularizer to avoid over-fitting. DyG2Vec is an efficient attention-based graph neural network that performs message-passing across structure and time to output task-agnostic node embeddings, without the need for expensive random-walk anonymyzation procedures (Wang et al., 2021b; Jin et al., 2022) or memory modules (Rossi et al., 2020; Souza et al., 2022). We equip DyG2Vec with the ability to perform non-contrastive SSL, which allows the model to learn rich representations without labels, if needed. Our results on 7 benchmark datasets indicate that on average, DyG2Vec outperforms the SoTA baseline CaW (Wang et al., 2021b) on the future link prediction task by 4.23% for the transductive setting and 3.30% for the inductive setting. In addition, DyG2Vec addresses the efficiency bottleneck often experienced with other dynamic graph encoding alternatives. It reduces the training/inference time by 5-10x compared to the state-of-the-art models, thereby providing superior model performance with a significantly reduced computational demand. This efficiency gain significantly enhances the model's scalability potential for large graphs.

Our main contributions can be summarized as follows:

- We propose an effective message-passing encoder that leverages temporal edge encoding to better capture temporal dependencies.

- We eliminate the need for memory modules or expensive causal random-walk extraction methods through efficient window-based subgraph encoding, making it easier to extract temporal motifs.

## 2 Related Work

We review the most relevant literature on dynamic graph and self-supervised representation learning. See Appendix A.6 for more tangentially related works.

**Representation learning for dynamic graphs:** Early works on representation learning for continuous-time dynamic graphs typically divide the graph into snapshots that are encoded by a static GNN and then processed by an RNN module (Sankar et al., 2020; Pareja et al., 2020; Kazemi et al., 2020). Such methods fail to learn fine-grained temporal patterns at smaller timescales within each snapshot. Therefore, several RNN-based methods were introduced that sequentially update node embeddings as new edges arrive. JODIE (Kumar et al., 2019) employs two RNN modules to update the source and destination embeddings of an arriving edge. DyRep (Trivedi et al., 2019) adds a temporal attention layer to take into account multi-hop interactions when updating node embeddings. TGAT (Xu et al., 2020) includes an Attention-based Message-Passing (AMP) architecture to aggregate messages from a historical neighborhood. TGN (Rossi et al., 2020) alleviates the expensive neighborhood aggregation of TGAT by using an RNN memory module to encode the history of each node. CaW (Wang et al., 2021b) extracts temporal patterns through an expensive procedure that samples temporal random walks and encodes them with an LSTM. This procedure must be performed for every prediction. PINT (Souza et al., 2022) is a memory-based method that leverages injective message-passing and relative positional encodings to overcome the theoretical weakness of both memory-based methods (e.g., TGN) and walk-based methods (e.g., CaW). Jin et al. (Jin et al., 2022) adapt CaW to

include spatio-temporal bias and exploitation-exploration trade-off sampling biases, employing differential equations (ODE) to effectively model the irregularly sampled temporal interactions of a node. NAT (Luo & Li, 2022) abandons the commonly used message-passing and walk-based paradigms and instead adopts dictionary-based learning by caching a fixed number of interactions for each node. Node representations are then built by aggregating temporal and structural features within the cache. Finally, GraphMixer (Cong et al., 2023b) uses a conceptually simple MLP-based link classifier that summarizes structural information from the latest temporal links and achieves surprisingly decent performance.

**Self-supervised representation learning:** Multiple works explore learning visual representations without labels (Liu et al., 2021). The more recent contrastive methods generate random views of images through data augmentations, and then force representations of positive pairs to be similar while pushing apart representations of negative pairs (Chen et al., 2020a; He et al., 2020). With the goal of attaining hard negative samples, such methods typically use large batch sizes Chen et al. (2020a) or memory banks (He et al., 2020; Chen et al., 2020b). Non-contrastive methods such as BYOL (Grill et al., 2020) and VICReg (Bardes et al., 2022) eliminate the need for negative samples through various techniques such as regularization or architecture tricks that avoid representation collapse (Jing et al., 2022). Recently, several SSL methods have been adapted to pre-train GNNs (Xie et al., 2022). BGRL (Thakoor et al., 2022) adapts BYOL to graphs to eliminate the need for negative samples, which are often memory-heavy in the graph setting. In this work, we follow a principled approach for SSL pre-training based on VICReg (Balestriero & LeCun, 2022) compared to other methods such as BGRL that rely on architecture tricks and heuristics.

Most adaptations of SSL for dynamic graphs have focused on improving downstream task performance via auxiliary losses rather than learning general pre-trained models. Previous works (Jiang et al., 2021) either use contrastive learning methods, which require high memory and computation due to negative sampling (Thakoor et al., 2022), or incorporate weak encoders (Tian et al., 2021), which leads to performance deterioration, particularly for large-scale graphs. Furthermore, readily adapting prior SSL methods to temporal domains is non-trivial as dynamic graphs can involve heavy distribution shifts. For example, new nodes arrive and others depart, and these arrival patterns occur at different timescales. As a result, there has been limited success in adapting SSL pre-training to dynamic graphs.

**Position of our work:** DyG2Vec relies on efficient message-passing GNNs without requiring the computationally expensive temporal causality on subgraph sampling (Xu et al., 2020). *Our architecture does not use complex memory-based architectures which require designing memory update schemes and can suffer from obsolete node memory for large batch sizes* (Zhou et al., 2022). While random-walk-based works (Wang et al., 2021b; Jin et al., 2022) alleviate these issues with online feature construction through causal walks, such methods are orders of magnitude slower and difficult to parallelize on GPUs (Luo & Li, 2022). **In contrast to prior works**, *our method neither maintains a cache or memory for each node nor requires the full history to make predictions.* Instead, it operates on a fixed-size window of the past relations to generate node embeddings. Furthermore, we fall under the message-passing paradigm which can leverage GPU parallelism using cutting-edge frameworks (Fey & Lenssen, 2019). Last, we propose a joint-embedding architecture that is compatible with recent SSL methods. In our experiments, we show how this allows the model to learn temporal patterns even without direct training on downstream tasks.

## 3 Problem formulation

A Continuous-Time Dynamic Graph (CTDG) $\mathcal{G} = (\mathcal{V}, \mathcal{E}, \mathcal{X})$ is a sequence of $E = |\mathcal{E}|$ interactions, where $\mathcal{X} = (X^V, X^E)$ is the set of input features containing the *node features* $X^V \in \mathbb{R}^{N \times D^1}$ and the *edge features* $X^E \in \mathbb{R}^{E \times D^2}$. $\mathcal{E} = \{e_1, e_2, \dots, e_E\}$ is the set of interactions. There are $N = |\mathcal{V}|$ nodes, and $D^1$ and $D^2$ are the dimensions of the node and edge feature vectors, respectively. An edge $e_i = (u_i, v_i, t_i, m_i)$ is an interaction between any two nodes $u_i, v_i \in \mathcal{V}$, with $t_i \in \mathbb{R}$ being a continuous timestamp, and $m_i \in X^E$ an edge feature vector. For simplicity, we assume that the edges are undirected and ordered by time (i.e., $t_i \leq t_{i+1}$). A temporal sub-graph $\mathcal{G}_{i,j}$ is defined as a set consisting of all the edges in the interval $[t_i, t_j]$, such that $\mathcal{E}_{ij} = \{e_k \mid t_i \leq t_k < t_j\}$. Any two nodes can interact multiple times throughout the time horizon; therefore, $\mathcal{G}$ is a multi-graph.

Our goal is to learn a model $f$ that maps the input graph to a representation space. The model is a pre-trainable encoder-decoder architecture, $f = (g_\theta, d_\gamma)$. The encoder $g_\theta$ maps a dynamic graph to node embeddings $\boldsymbol{H} \in \mathbb{R}^{N \times D^1}$; the decoder $d_\gamma$ performs a task-specific prediction given the embeddings. The model is parameterized by the encoder/decoder parameters $(\theta, \gamma)$. More concretely,

$$\boldsymbol{H} = g_\theta(\mathcal{G}), \qquad \boldsymbol{z} = d_\gamma(\boldsymbol{H}; \bar{e}), \tag{1}$$

where $\boldsymbol{z} \in \mathbb{R}^{D^Y}$ is the prediction of task-specific labels (e.g., edge prediction or source node classification labels) of the target (future) edge $\bar{e}$. The node embeddings $\boldsymbol{H}$ must capture the temporal and structural dynamics of each node such that the future can be accurately predicted from the past, e.g., future edge prediction given past edges. The main distinction of this design is that, unlike previous dynamic graph models (Rossi et al., 2020; Xu et al., 2020; Wang et al., 2021b), the encoder must produce embeddings independent of the downstream task specifications. This special trait can allow the model to be compatible with the SSL paradigm where an encoder is pre-trained separately and then fine-tuned together with a task-specific decoder to predict the labels.

To this end, we present a novel DyG2Vec framework, that can learn rich node embeddings at any timestamp $t$ independent of the downstream task. DyG2Vec is formulated as a two-stage framework. In the first stage, we use a non-contrastive SSL method to learn the model $f^{SSL} = (g_\theta, d_\psi)$ over various sampled dynamic sub-graphs with self-supervision. $d_\psi$ is an SSL decoder that is only used in the SSL pre-training stage. In the second stage, a task-specific decoder $d_\gamma$ is trained on top of the pre-trained encoder $g_\theta$ to compute the outputs for the downstream tasks, e.g., future edge prediction or dynamic node classification (Xu et al., 2020; Wang et al., 2021b).

We consider two example downstream tasks: future link prediction (FLP), and dynamic node classification (DNC). In each task, we make a prediction on a set of target (positive) edges $\bar{\mathcal{E}}$. For FLP, this is augmented by a set of negative edges. Each negative edge $(u_j, v'_j, t_j, m_j)$ differs from its corresponding positive edge only in the destination node, $v'_j \neq v_j$, which is selected at random from all nodes. The FLP task is then binary classification for the test set of $2|\bar{\mathcal{E}}|$ edges. In the DNC task, a dynamic label is associated with each node that participates in an interaction. We are provided with $\{(u_j, t_j)\}$, i.e., the source node and interaction time. The goal is to predict the source node labels for the test interactions. *It is important to note that each prediction must be made given only access to the past, i.e., edges before time $t_j$.* The performance metrics are detailed in Appendix A.3.

## 4 Methodology

We now introduce our novel dynamic graph learning framework DyG2Vec. We first outline the encoder architecture. We then introduce the window-based downstream training approach. Finally, we present the SSL approach that can be *optionally* used to pre-train the dynamic graph encoder.

### 4.1 DyG2Vec Encoding Model

Our encoder combines a self-attention mechanism for message-passing with a learnable time-encoding module that provides relative time encoding. We also introduce a novel temporal edge encoding that efficiently captures the temporal structural relationship between nodes. The full architecture is outlined in Figure 1. For simplicity, we define the message passing on any input graph $\mathcal{G}$; however, as shown in Figure 1, the input graph is restricted to be a window of the full graph. See Section 4.2 for details.

**Temporal Attention Embedding**: Given any input dynamic graph $\mathcal{G}$, the encoder $g_\theta$ computes the embedding $\boldsymbol{h}_i^L \in \mathbb{R}^{D^1}$ of node $i$ through a series of $L$ multi-head attention (MHA) layers (Vaswani et al., 2017) that aggregate messages from its $L$-hop neighborhood (Xu et al., 2020; Velickovic et al., 2018).

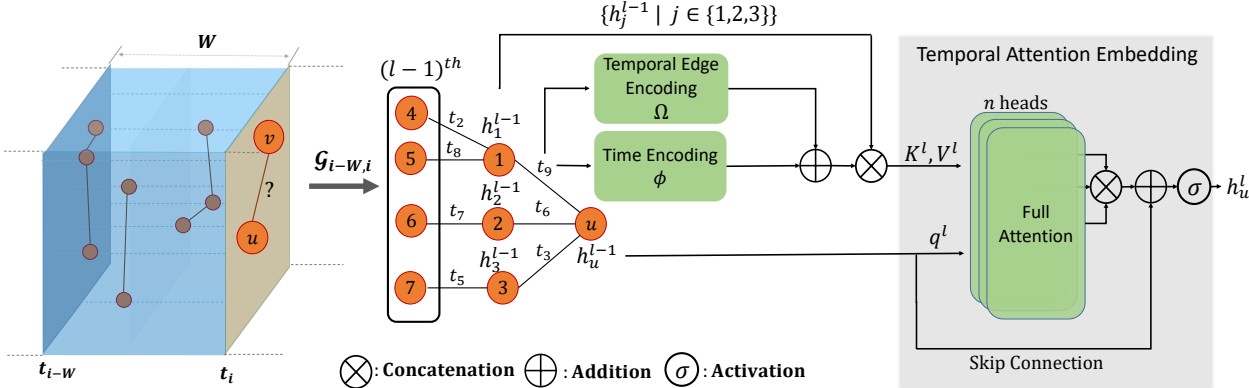

Figure 1: Using DyG2Vec window framework to encode the target node $u$. Every slice of the dynamic graph $\mathcal{G}$ contains edges that arrived at the same continuous timestamp. The blue interval represents the history graph $\mathcal{G}_{i-W,i}$ that is encoded to make a prediction on the target edge $(u, v)$. Note that both $u$ and $v$ share the same sampled history graph. For simplicity, we omit edge features $m_p$ from the attention encoder.

Given a node embedding $\boldsymbol{h}_i^{l-1}$ at layer $l-1$, we uniformly sample $N$ 1-hop neighborhood interactions of node $i$, $\mathcal{N}(i) = \{e_p, \dots, e_k\} \subseteq \mathcal{E}$, where $p, k \in \{1, |\mathcal{E}|\}$. The embedding $\mathbf{h}_i^l$ at layer $l$ is calculated by:

$$\mathbf{h}_i^l = \mathbf{W}_1 \mathbf{h}_i^{l-1} + \texttt{MHA}^l(\mathbf{q}^l, \mathbf{K}^l, \mathbf{V}^l), \tag{2}$$

$$\mathbf{q}^l = \mathbf{h}_i^{l-1}, \tag{3}$$

$$\mathbf{K}^l = \mathbf{V}^l = [\Phi_p^{l-1}(t_p), \dots, \Phi_k^{l-1}(t_k)]. \tag{4}$$

Here, $\mathbf{W}_1$ is a learnable mapping matrix, $\texttt{MHA}^l(\cdot)$ is a multi-head dot-product attention layer, and $\Phi_p^{l-1}(t_p)$ represents the edge feature vector of edge $e_p = (u_p, v_p, t_p, \boldsymbol{m}_p) \in \mathcal{N}(i)$ at time $t_p$:

$$\Phi_p^{l-1}(t_p) = [\boldsymbol{h}_{u_p}^{l-1} \,||\, \boldsymbol{f}_p(t_p) \,||\, \boldsymbol{m}_p], \tag{5}$$

$$\boldsymbol{f}_p(t_p) = \phi(\bar{t}_i - t_p) + \Omega_p(t_p), \tag{6}$$

$$\bar{t}_i = \max\{t_l \mid e_l \in \mathcal{N}(i)\}, \tag{7}$$

where $||$ denotes concatenation and the **Time Encoding** module $\phi(t) = [\cos\omega_1 t, \dots, \cos\omega_{D^1} t]$ is a learnable Time2Vec module that helps the model be aware of the relative timespan between a sampled interaction and the most recent interaction of node $i$ in the input graph. $\Omega_p(.) \in \mathbb{R}^{D^1}$ is a temporal edge encoding function, described in more detail below. In contrast to TGAT's recursive message passing procedure (Xu et al., 2020), the message passing in our encoder is 'flat': at every iteration, the same set of node embeddings is used to propagate messages to neighbors. That is, we do not restrict messages to flow towards the source node only but rather treat the *sampled temporal graph as undirected*. This allows the encoder to better capture the multi-hop common neighbors between the target nodes, which are vital to learning the temporal motifs and predicting future interactions. Moreover, unlike CaW (Wang et al., 2021b), *we do not restrict the neighbor sampling to go backwards in time* (i.e. causal sampling) as we found this to be too restrictive and degrade the overall performance on downstream tasks (See Section 6). Lastly, note that the relative time encoding is with respect to the latest timestamp, $\bar{t}_i$, incident to the source and not with respect to the target edge timestamp; hence, allowing the encoding step to be independent of the prediction (decoding) step and making the generated embeddings task-agnostic.

**Temporal Edge Encoding**: Dynamic graphs often follow evolutionary patterns that reflect how nodes interact over time (Kovanen et al., 2011). For example, in social networks, two people who share many friends are likely to interact in the future. Therefore, we incorporate two simple yet effective temporal encoding methods that provide inductive biases to capture common structural and temporal evolutionary

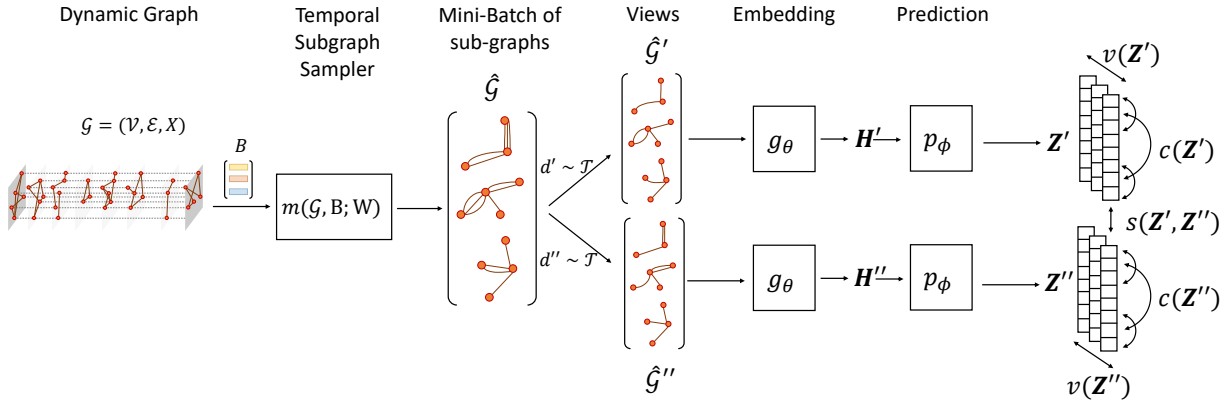

Figure 2: The joint embedding architecture for the non-contrastive SSL Framework. Each slice of the input dynamic graph contains edges arriving at the same continuous timestamp. $B$ is a batch of intervals of size $W$. $\hat{\mathcal{G}}$ is a batch of the corresponding input graphs of each interval.

behaviour of dynamic graphs. The temporal edge encoding function, parameterized by $W \in \mathbb{R}^{D^1 \times 3}$, is then:

$$\Omega_p(t_p) = \mathbf{W}_2[z_p(t_p)||c_p(t_p)], \tag{8}$$

where we incorporate (i) *Temporal Degree Centrality* $z_p(t_p) \in \mathbb{R}^2$: the concatenated current degrees of nodes $u_p$ and $v_p$ at time $t_p$; and (ii) *Common Neighbors* $c_p(t_p) \in \mathbb{R}$: the number of common 1-hop neighbors between nodes $u_p$ and $v_p$ at time $t_p$.

By using the degree centrality as an edge feature, the model is able to learn any bias towards more frequent interactions with high-degree nodes. The number of common neighbors helps capture temporal motifs, and it is known to often have a strong positive correlation with the likelihood of a future interaction (Yao et al., 2016).

## 4.2 DyG2Vec Downstream Training

In the downstream training stage, the DyG2Vec model $f = (g_\theta, d_\gamma)$ consists of the encoder $g_\theta$ and a task-specific decoder $d_\gamma$ which is trained using a similar window-based training strategy. The model is trained to make predictions depending on the downstream tasks (e.g., link prediction or node classification). It is important to note that all tasks considered for dynamic graphs involve predicting a (future) target edge given access to the past interactions. However, rather than having access to all past edges, we limit the model to a fixed window of $W$ interactions. That is, to predict a target edge $\bar{e} = (u_j, v_j, t_j, m_j)$, we sample an input (history) graph $\mathcal{G}_{j-W,j}$ from the time interval $\{t_{j-W}, t_j\}$, centered at $u_j$ and $v_j$, and make a prediction as follows: $\boldsymbol{H} = g_\theta(\mathcal{G}_{j-W,j})$ is the matrix of node embeddings returned by the encoder, and $\boldsymbol{z} = d_\gamma(\boldsymbol{H}; \bar{e})$ is the prediction output of the decoder. The model parameters are optimized by training with a loss function $\mathcal{L}_D(\boldsymbol{z}, \boldsymbol{o})$, where $\mathcal{L}_D$ is defined depending on the downstream task and $\boldsymbol{o}$ contains task-specific labels (See Section 3). It is important to note that, unlike previous methods (Xu et al., 2020; Wang et al., 2021b), the *embeddings of $u_j$ and $v_j$ are generated through message passing on the same sampled graph.* Consequently, the encoder can better recognize similar historical patterns between the target nodes without the need for costly motif-correlation through counting that is performed in walk-based methods (Wang et al., 2021b; Jin et al., 2022).

The window-based training strategy has several major advantages. First, the window acts as a regularizer by providing a natural inductive bias towards recent edges, which are often more predictive of the immediate future. Second, it avoids costly time-based neighborhood sampling (Wang et al., 2021b). Third, relying on a fixed window-size for message-passing allows for constant memory and computational complexity, which is well-suited to the practical *online streaming* data scenario.

### 4.3 Self-supervised Pre-training for Dynamic Graphs

Previous work (Paranjape et al., 2017) has shown that temporal motifs develop at different timescales throughout a dynamic graph. For example, question-answer patterns on StackOverflow typically take 30 min to develop while messaging patterns on social media platforms can take less than 20 minutes to form. Inspired by such observations, we outline a window-based pre-training strategy where the encoder can be pre-trained on a sliding window of the dynamic graph in an effort to learn the fine-grained temporal patterns throughout the time horizon. The full pre-training procedure is displayed in Figure 2.

Given the full input dynamic graph $\mathcal{G}_{0,E}$, a set of intervals $I$ is generated by dividing the entire time-span $\{t_0, t_E\}$ into $M = \lceil E/S \rceil - 1$ intervals with stride $S$ and interval length $W$ (See Appendix A.3 for details). Let $B \subset I$ be a mini-batch (randomly sampled subset) of intervals. Given $B$, the *temporal subgraph sampler* $m(\mathcal{G}, B; W)$ constructs the mini-batch of input graphs: $\hat{\mathcal{G}} = \{\mathcal{G}_{i,j} \mid [i,j] \in B\}$. In principle, $\mathcal{G}_{i,j} \in \hat{\mathcal{G}}$ is an input graph to the SSL pre-training. The parameter $W$ controls the size of the window while $S$ controls the stride between intervals. In practice, we found that setting $S = 200$ and $W = 32K$ gives a reasonable trade-off to learn both the long-range and short-range patterns within the dynamic graph.

We formulate a joint-embedding architecture (Bromley et al., 1993) for DyG2Vec in which two views of a mini-batch of sub-graphs are generated through random transformations. The transformations are randomly sampled from a distribution defined by a distortion pipeline. The encoder maps the views to node embeddings which are processed by the predictor to generate node representations. We minimize an SSL objective (Eq. 9, described below) to optimize the model parameters end-to-end in the pre-training stage.

**Views**: The temporal distortion module generates two views of the input graphs $\hat{\mathcal{G}}' = d'(\hat{\mathcal{G}})$ and $\hat{\mathcal{G}}'' = d''(\hat{\mathcal{G}})$ where the transformations $d'$ and $d''$ are sampled from a distribution $\mathcal{T}$ over a pre-defined set of candidate graph transformations. In this work, we use edge dropout and edge feature masking (Thakoor et al., 2022) in the transformation pipeline. See Appendix A.3 for more details.

**Embedding**: The encoding model $g_\theta$ is an Attention-based Message-Passing (AMP) neural network presented in Sec. 4.1. It produces node embeddings $\boldsymbol{H}'$ and $\boldsymbol{H}''$ for the views $\hat{\mathcal{G}}'$ and $\hat{\mathcal{G}}''$ of the input graphs $\mathcal{G}_{i,j}$. We elaborate on the details of the encoder in Sec. 4.1.

**Prediction**: The decoding head $d_\psi$ for our self-supervised learning design consists of a MLP predictor $p_\phi$ that outputs the final representations $\boldsymbol{Z}'$ and $\boldsymbol{Z}''$, where $\boldsymbol{Z} = p_\phi(\boldsymbol{H})$.

**SSL Objective**: In order to learn useful representations, we minimize a regularization-based SSL loss function (Bardes et al., 2022):

$$\mathcal{L}^{SSL} = \lambda s(\boldsymbol{Z}', \boldsymbol{Z}'') + \mu[v(\boldsymbol{Z}') + v(\boldsymbol{Z}'')] + \nu[c(\boldsymbol{Z}') + c(\boldsymbol{Z}'')]. \tag{9}$$

In this loss function, the weights $\lambda$, $\mu$, and $\nu$ control the emphasis placed on each of three regularization terms. The *invariance* term $s$ encourages representations of the two views to be similar. The *variance* term $v$ is included to prevent the well-known collapse problem (Jing et al., 2022). The covariance term $c$ promotes maximization of the information content of the representations. See Appendix A.1 for details.

Note that this pre-training stage is only performed when needed i.e. target labels are scarce. Following the pre-training stage, we replace the SSL decoder with a task-specific downstream decoder $d_\gamma$ that is trained on top of the *frozen* pre-trained encoder.

## 5 Experimental Evaluation

### 5.1 Experimental Setup

**Baselines**: We compare DyG2Vec to five state-of-the-art baseline models: DyRep (Trivedi et al., 2019), JODIE (Kumar et al., 2019), TGAT (Xu et al., 2020), TGN (Rossi et al., 2020), CaW (Wang et al., 2021b), and NAT (Luo & Li, 2022). DyRep, JODIE, and TGN sequentially update node embeddings using an RNN. TGAT applies message passing via attention on a sampled temporal subgraph. CaW samples temporal random walks and learns temporal motifs by counting node occurrences in each walk. NAT builds temporal

Table 1: Dynamic Graph Datasets. **% Repetitive Edges**: % of edges which appear more than once.

| Dataset | # Nodes | # Edges | # Unique Edges | Edge Features | Node Labels | Bipartite | % Repetitive Edges |
|---|---|---|---|---|---|---|---|
| Reddit | 11,000 | 672,447 | 78,516 | ✓ | ✓ | ✓ | 54% |
| Wikipedia | 9,227 | 157,474 | 18,257 | ✓ | ✓ | ✓ | 48% |
| MOOC | 7,144 | 411,749 | 178,443 | ✓ | ✓ | ✓ | 53% |
| LastFM | 1980 | 1,293,103 | 154,993 | | | ✓ | 68% |
| UCI | 1899 | 59,835 | 13838 | | | ✓ | 62% |
| Enron | 184 | 125,235 | 2215 | | | | 92% |
| SocialEvolution | 74 | 2,099,519 | 2506 | | | | 97% |

node representations using a cache that stores a limited set of historical interactions for each node. Appendix A.2.3 contains additional comparisons to the GraphMixer (Cong et al., 2023b) baseline.

Table 2: Future link Prediction Performance in AP (Mean $\pm$ Std). **Bold** font and ul font represent first- and second-best performance respectively. DyG2Vec is trained end-to-end with no pre-training.

| Setting | Model | Wikipedia | Reddit | MOOC | LastFM | Enron | UCI | SocialEvol. |
|---|---|---|---|---|---|---|---|---|
| Transductive | JODIE | $0.956 \pm 0.002$ | $0.979 \pm 0.001$ | $0.797 \pm 0.01$ | $0.691 \pm 0.010$ | $0.785 \pm 0.020$ | $0.869 \pm 0.010$ | $0.847 \pm 0.014$ |
| | DyRep | $0.955 \pm 0.004$ | $0.981 \pm 1e\text{-}4$ | $0.840 \pm 0.004$ | $0.683 \pm 0.033$ | $0.795 \pm 0.042$ | $0.524 \pm 0.076$ | $0.885 \pm 0.004$ |
| | TGAT | $0.968 \pm 0.001$ | $0.986 \pm 3e\text{-}4$ | $0.793 \pm 0.006$ | $0.633 \pm 0.002$ | $0.637 \pm 0.002$ | $0.835 \pm 0.003$ | $0.631 \pm 0.001$ |
| | TGN | $0.986 \pm 0.001$ | $0.985 \pm 0.001$ | $0.911 \pm 0.010$ | $0.743 \pm 0.030$ | $0.866 \pm 0.006$ | $0.843 \pm 0.090$ | $0.966 \pm 0.001$ |
| | CaW | $0.976 \pm 0.007$ | $0.988 \pm 2e\text{-}4$ | $0.940 \pm 0.014$ | $0.903 \pm 1e\text{-}4$ | $0.970 \pm 0.001$ | $0.939 \pm 0.008$ | $0.947 \pm 1e\text{-}4$ |
| | NAT | $0.987 \pm 0.001$ | $0.991 \pm 0.001$ | $0.874 \pm 0.004$ | $0.859 \pm 1e\text{-}4$ | $0.924 \pm 0.001$ | $0.944 \pm 0.002$ | $0.944 \pm 0.010$ |
| | **DyG2Vec** | **$0.995 \pm 0.003$** | **$0.996 \pm 2e\text{-}4$** | **$0.980 \pm 0.002$** | **$0.960 \pm 1e\text{-}4$** | **$0.991 \pm 0.001$** | **$0.988 \pm 0.007$** | **$0.987 \pm 2e\text{-}4$** |
| Inductive | JODIE | $0.891 \pm 0.014$ | $0.865 \pm 0.021$ | $0.707 \pm 0.029$ | $0.865 \pm 0.03$ | $0.747 \pm 0.041$ | $0.753 \pm 0.011$ | $0.791 \pm 0.031$ |
| | DyRep | $0.890 \pm 0.002$ | $0.921 \pm 0.003$ | $0.723 \pm 0.009$ | $0.869 \pm 0.015$ | $0.666 \pm 0.059$ | $0.437 \pm 0.021$ | $0.904 \pm 3e\text{-}4$ |
| | TGAT | $0.954 \pm 0.001$ | $0.979 \pm 0.001$ | $0.805 \pm 0.006$ | $0.644 \pm 0.002$ | $0.693 \pm 0.004$ | $0.820 \pm 0.005$ | $0.632 \pm 0.005$ |
| | TGN | $0.974 \pm 0.001$ | $0.954 \pm 0.002$ | $0.855 \pm 0.014$ | $0.789 \pm 0.050$ | $0.746 \pm 0.013$ | $0.791 \pm 0.057$ | $0.904 \pm 0.023$ |
| | CaW | $0.977 \pm 0.006$ | $0.984 \pm 2e\text{-}4$ | $0.933 \pm 0.014$ | $0.890 \pm 0.001$ | $0.962 \pm 0.001$ | $0.931 \pm 0.002$ | $0.950 \pm 1e\text{-}4$ |
| | NAT | $0.986 \pm 0.001$ | $0.986 \pm 0.002$ | $0.832 \pm 1e\text{-}4$ | $0.878 \pm 0.003$ | $0.949 \pm 0.010$ | $0.926 \pm 0.010$ | $0.952 \pm 0.006$ |
| | **DyG2Vec** | **$0.992 \pm 0.001$** | **$0.991 \pm 0.002$** | **$0.938 \pm 0.010$** | **$0.979 \pm 0.006$** | **$0.987 \pm 0.004$** | **$0.976 \pm 0.002$** | **$0.978 \pm 0.010$** |

**Downstream Tasks**: We evaluate all models on two temporal tasks: future link prediction (FLP), and dynamic node classification (DNC). In FLP, the goal is to predict the probability of future edges occurring given the source, destination, and timestamp. For each positive edge, we sample a negative edge that the model is trained to predict as negative. The DNC task involves predicting the label of the source node of a future interaction. Both tasks are trained using binary cross entropy loss. For FLP, we evaluate all models on the transductive and inductive settings. The latter is a more challenging setting where a model makes a prediction on unseen nodes. See Appendix A.3 for details.

For the FLP task, we report the Average Precision (AP) metric. For the DNC task, we report the area under the curve (AUC) metric due to the prevailing issue of class imbalance in dynamic graphs. *By default, DyG2Vec is trained end-to-end with no pre-training, just like the baselines.* The pre-training stage is only performed to test the potential benefits of SSL (e.g. Fig. 4). Additional results on the benefits of SSL can be found in Appendix A.2.

**Datasets**: We use 7 real-world datasets: Wikipedia, Reddit, MOOC, and LastFM (Kumar et al., 2019); SocialEvolution, Enron, and UCI (Wang et al., 2021b). These datasets span a wide range in terms of number of nodes and interactions, time range, and repetition ratio. The dataset statistics are presented in Table 1. We perform the same 70%-15%-15% chronological split for all datasets as in Wang et al. (2021b). The datasets are split differently under two settings: Transductive and Inductive. Under the transductive setting, a dataset is split normally by time, i.e., the model is trained on the first 70% of links and tested on the rest. In the inductive setting, we strive to test the model's prediction performance on edges with unseen nodes. Therefore, following Wang et al. (2021b), we randomly assign 10% of the nodes to the validation and test sets and remove any interactions involving them in the training set. Additionally, to ensure an inductive setting, we remove any interactions not involving these nodes from the test set.

**Training Protocols and Hyperparameters**: In Table 2, DyG2Vec is initialized with random parameters and trained end-to-end on the downstream tasks and compared to all supervised baselines. In the semi-supervised evaluation setting (left figure of Fig. 4), the decoder is trained on top of the frozen (SSL pre-trained) encoder on a random portion of the dataset (i.e., a fraction of the target edges). The DyG2Vec encoder performs $L = 3$ layers of message passing. We sample $N = 64$ temporal neighbors at the first hop and 1 neighbor at the second and third hops. All neighbors are sampled uniformly at random. We found that uniform sampling within a window works better than only looking at the latest $N$ neighbors of a node (Xu et al., 2020; Rossi et al., 2020). Other hyperparameters are discussed in Appendix A.3. For the DNC task, following prior work (Rossi et al., 2020), the decoder is trained on top of the frozen encoder that is pre-trained on the future link prediction task.

## 5.2 Experimental Results

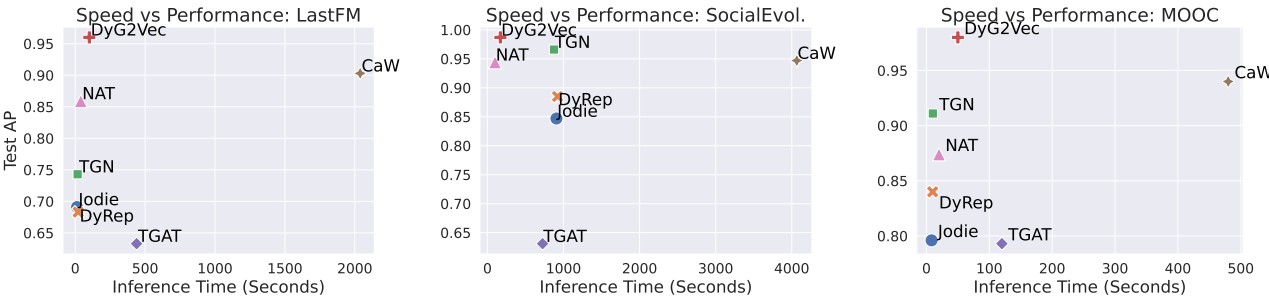

Figure 3: Transductive FLP Performance (Test AP) vs Inference runtime (s) on 3 datasets. Inference time represents the time it takes to predict the whole test set. The test sets are approximately of size 400K, 600K, and 100K edges respectively.

**Future Link Prediction**: We report the test AP scores for future link prediction in Table 2. Our model outperforms all sequential and message-passing baselines on 7/7 of the datasets in the transductive setting. The gap is particularly large on the UCI and LastFM datasets, where DyG2Vec outperforms the second-best methods (NAT and CaW) by over 4% and 6% respectively. Interestingly, while SocialEvol. is the largest dataset with $\sim 2M$ edges, our model is able to achieve SoTA performance while only using the last 8000 edges to predict any future edge. This further cements the findings in Xu et al. (2020) that capturing recent interactions may be more important for certain tasks. Our window-based framework offers a good trade-off between capturing recent interactions and recurrent patterns which both have a major influence on future interactions. In the inductive settings, most methods drop in performance due to the difficult nature of predicting over unseen nodes. However, DyG2Vec still outperforms the best methods significantly (e.g., 8% gap for LastFM) which demonstrates its ability to learn temporal motifs rather than overfitting to node identities.

Table 3: Transductive Dynamic Node Classification Performance in AUC (Mean ± Std). Avg. Rank reports the mean rank of a method across all datasets.

| Model | Wikipedia | Reddit | MOOC | Avg. Rank ↓ |
|---|---|---|---|---|
| TGAT | $0.800 \pm 0.010$ | $\mathbf{0.664 \pm 0.009}$ | $0.673 \pm 0.006$ | 3.0 |
| JODIE | $0.843 \pm 0.003$ | $0.566 \pm 0.016$ | $0.672 \pm 0.002$ | 3.7 |
| Dyrep | $\mathbf{0.873 \pm 0.002}$ | $0.633 \pm 0.008$ | $0.661 \pm 0.012$ | 3.3 |
| TGN | $0.828 \pm 0.004$ | $\underline{0.655 \pm 0.009}$ | $\underline{0.674 \pm 0.007}$ | 2.3 |
| **DyG2Vec** | $0.824 \pm 0.050$ | $0.649 \pm 0.020$ | $\mathbf{0.785 \pm 0.005}$ | 2.6 |

**Dynamic Node classification**: We evaluate DyG2Vec on 3 datasets for node classification where the labels indicate whether a user will be banned from editing/posting after an interaction. This task is challenging both due to its dynamic nature (i.e., nodes can change labels) and the high class imbalance (only 217 of

157K interactions result in a ban). We measure performance using the AUC metric to deal with the class imbalance. Table 3 shows that DyG2Vec outperforms all baselines on the MOOC dataset significantly by over 10%. For Wikipedia and Reddit, DyG2Vec is within $2 - 5\%$ of the best performance. Overall, none of the methods display the best performance consistently across all 3 datasets. We believe this is due to the high class imbalance problem which makes it a better fit for anomaly detection methods (Ranshous et al., 2015).

**Training/Inference Speed**: Relying on a fixed window of history to produce task-agnostic node embeddings gives DyG2Vec a significant advantage in speed and memory. Figures 3 and 6 show the performance and runtime per epoch of all methods on the three large datasets: LastFM, SocialEvolution and MOOC. DyG2Vec is many orders of magnitude faster than CaW due to the latter's expensive random walk sampling procedure. RNN-based methods such as TGN have a good runtime on LastFM and MOOC; however, they are significantly slower on SocialEvol. which has a small number of nodes (74) but a large number of interactions ($\sim 2M$). This suggests that memory-based methods are slower for settings where a node's memory is updated frequently. Furthermore, while TGAT has a similar AMP encoder, DyG2Vec improves the efficiency and performance significantly. This reveals the significance of the window-based mechanism and the encoder architecture. Overall, DyG2Vec presents the best trade-off between speed and performance. A more detailed complexity analysis is included in Appendix A.2.

**Semi-supervised Learning on Dynamic Node Classification**: The DNC task is challenging due to its highly imbalanced labels. In Figure 4, we show that SSL is an effective pre-training strategy for the DNC task, particularly in the low-label data regime where each model is trained on a portion of the target edges. This highlights the potential of SSL to effectively use unlabeled data for representation learning and prevent representations from overfitting to such imbalanced classification tasks.

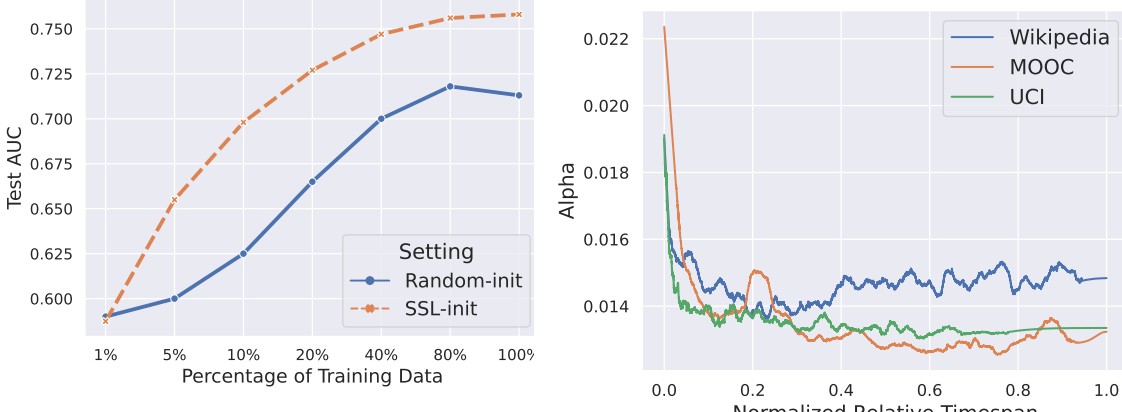

Figure 4: First figure plots Semi-Supervised Learning results on Dynamic Node Classification. For each setting, DyG2Vec was trained on a varying random portion of the training data. Second figure plots the Average Attention Weight versus Relative Timespan for DyG2Vec trained with $W = 64K$. The relative timespan is normalized with the maximum timespan across all interactions. A higher timespan means a farther interaction.

## 5.3   Ablation and Sensitivity Analysis

We perform a detailed study on different instances of our framework with 3 datasets. All ablation results are reported in Figure 5.

**Window Size**: We observe that a large window size works best for most datasets. However, we see a minor drop in performance ($\sim 1\%$) for MOOC due to the inherently different recurring temporal patterns. As observed by Xu et al. (2020), recent and/or recurrent interactions are often the most predictive of future interactions. Therefore, datasets with long range dependencies favor larger window sizes to capture the recurrent patterns while some datasets benefit from an increased bias towards recent interactions. Our window-based framework coupled with uniform neighbor sampling strikes a balance between the two. This

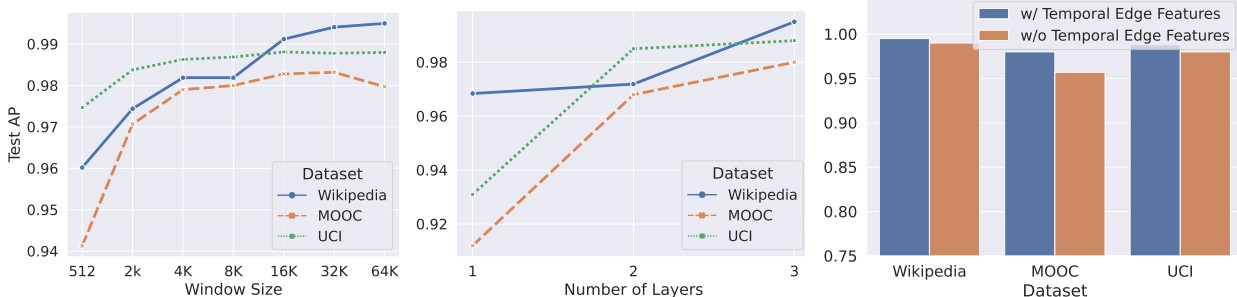

Figure 5: Ablation, sensitivity, and attention analysis on 3 datasets for the FLP transductive task.

shows that the fixed window size also contributes to the performance as it helps limit irrelevant information that is not highly predictive of future interactions. Nonetheless, as we show in Section 6, the attention-based encoder coupled with the time encoding function is able to learn the innate temporal dependencies regardless of the window size.

**Number of Layers:** Increasing the number of embedding layers improves performance for most datasets, and this effect is more noticeable for some (e.g., MOOC). This suggests that these datasets contain higher order temporal correlations among the nodes that must be learned using long-range message passing. Overall, the results show that one can choose to sacrifice some performance to further improve the speed of DyG2Vec by decreasing the window size and the number of layers.

**Temporal Edge Features**: The results show a substantial decrease in performance for MOOC when temporal edge features are removed (i.e., 1-4% drop). This indicates that such temporal edge features provide useful multi-hop information about the evolution of the dynamic graph (Yao et al., 2016).

# 6 Analysis

**Effect of Encoder Architecture**: As noted earlier, one of the main advantages of DyG2Vec over the baselines is its transformer-based architecture. Combined with the simple time-encoding module, DyG2Vec can selectively aggregate temporal and structural information across neighbors that are relevant to the target prediction. Moreover, unlike the baselines, message passing for the source and target nodes is performed on a shared undirected sampled subgraph to easily detect common neighbors across time which are vital to predicting future interactions. In Table 4, we show the effect of replacing our encoder architecture with the MPNN model (Gilmer et al., 2017) which performs simple message passing with our temporal edge encodings. The results show significant degradation (4-9%), displaying the benefits of transformers in attending to relevant history.

Table 4: Benefits of the encoder architecture in the FLP task. In the last row, we replace the transformer architecture (Sec. 4.1) with the simple MPNN architecture (Gilmer et al., 2017). All experiments are performed for one trial only in the transductive setting.

| Model | UCI | LastFM | Enron | MOOC |
|---|---|---|---|---|
| DyG2Vec (Default) | **0.981** | **0.960** | **0.990** | **0.988** |
| DyG2Vec with MPNN | 0.942 | 0.921 | 0.907 | 0.916 |

**Attention Weight Analysis**: A particular advantage of an attention-based architecture is that attention weights allow for easy interpretability of the learned temporal dependencies. In Figure 4 (right plot), we plot the attention weights $\alpha_{ij}$ with respect to the relative timespan. That is, for each test target edge $e_i = (u_i, v_i, t_i)$, we plot the attention weights to the one-hop neighbors of both target nodes, $\{\alpha_{u_i v_p} | v_p \in \mathcal{N}(u_i)\} \cup \{\alpha_{v_i v_k} | v_k \in \mathcal{N}(v_i)\}$, versus the relative timespans, $\{\bar{t}_{u_i} - t_p\} \cup \{\bar{t}_{v_i} - t_k\}$. Here, $\bar{t}_{u_i}$ represents the maximum timestamp incident to node $u_i$. Therefore, the attention weights indicate how much the model is attending to old and recent interactions. Unsurprisingly, higher importance is given to the most recent

interactions. However, both Wikipedia and UCI display higher weights for larger timespans compared to MOOC, indicating that they have long-range dependencies. This explains why performance monotonically increases for Wikipedia as $W$ increases while slightly degrades for the MOOC dataset, as seen in Figure 5.

**Neighbor Sampling and Temporal Edge Encoding**: In Table 5, we study the effect of neighbor sampling and temporal edge encoding on the downstream FLP task. Although Fig. 5 shows the importance of multi-hop sampling for discovering high-order temporal motifs, we found that DyG2Vec gives more importance to one-hop neighbors for most datasets. In fact, sampling 64 one hop neighbors gives SoTA performance compared to sampling 20 neighbors per hop (see $1^{st}$ and $5^{th}$ rows in Table 5). This suggests that the 1-hop recent interactions within a window are the most representative interactions for future prediction tasks. Moreover, unlike prior random-walk and AMP methods (Xu et al., 2020; Jin et al., 2022; Wang et al., 2021b), which argue for causal sampling (i.e. sampling backwards in time) to discover evolving temporal motifs, we have found this form of sampling to have little effect on the performance (See $2^{nd}$ row). Lastly, removing edge encodings almost always hurts performance. In fact, performing causal sampling with 20 neighbors at each hop, as done in TGAT, and removing temporal edge encodings causes up to 8% drop in performance (See last row). Additional analysis on the effect of temporal edge encodings on baselines can be found in Appendix A.2.

Table 5: Effect of neighbor sampling and temporal edge encoding on performance. The first row is the default setting where we sample 64,1,1 neighbors at the first, second and third hops respectively.

| Temporal Edge Encoding | Causal Sampling | Num Neighbors | Wikipedia | MOOC | UCI |
|:---:|:---:|:---:|:---:|:---:|:---:|
| ✓ | | 64,1,1 | **0.995** | 0.982 | **0.988** |
| ✓ | ✓ | 64,1,1 | 0.993 | **0.984** | 0.986 |
| | | 64,1,1 | 0.990 | 0.957 | 0.980 |
| | ✓ | 64,1,1 | 0.989 | 0.965 | 0.976 |
| ✓ | | 20,20,20 | 0.992 | 0.949 | 0.981 |
| ✓ | ✓ | 20,20,20 | 0.984 | 0.955 | 0.971 |
| | | 20,20,20 | 0.990 | 0.927 | 0.958 |
| | ✓ | 20,20,20 | 0.982 | 0.906 | 0.946 |

## 7   Conclusion

We introduce DyG2Vec, a novel window-based encoder-decoder model for dynamic graphs. It is an efficient attention-based message-passing model that utilizes multi-head attention modules to encode node embeddings across time. Furthermore, we present a joint-embedding architecture for dynamic graphs in which two views of temporal sub-graphs are encoded to minimize a non-contrastive loss function. Our window-based architecture allows for efficient message-passing and robust prediction abilities. We aim to further explore ways to improve the capacity of the dynamic graph models to learn long-range dependencies. Additionally, it seems promising to investigate other SSL paradigms aligned with temporal graphs.

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

# A Appendix

## A.1 Preliminary: VICReg

We outline the details of the VICReg (Bardes et al., 2022) method used in our SSL pre-training stage. Given the representations of two random views of an object (e.g., image) generated through random distortions, the objective of non-contrastive SSL is two-fold. First, the output representation of one view should be maximally informative of the input representation of the view. Second, the representation of one view should be maximally predictable from the representation of the other view. These two aspects are formulated by VICReg (Bardes et al., 2022) where a combination of 3 loss terms (i.e. Variance, Covariance, Invariance) is minimized to learn useful representations while also avoiding the well-known problem of collapse (Jing et al., 2022). More concretely, let $\boldsymbol{Z}' = [\boldsymbol{z}'_1, \ldots, \boldsymbol{z}'_n]$ and $\boldsymbol{Z}'' = [\boldsymbol{z}''_1, \ldots, \boldsymbol{z}''_n]$ be the batches composed of $n$ representations of dimension $d$.

**Variance term**: The variance regularization term $v$ is the mean over the representation dimension of the hinge function on the standard deviation of the representations along the batch dimension: $v(\boldsymbol{Z}) = \frac{1}{D} \sum_{j=1}^{D} \max(0, \gamma - S(\boldsymbol{Z}_{:,j}, \epsilon))$. Here $\boldsymbol{Z}_{:,j}$ is the column $j$ of matrix $\boldsymbol{Z}$. $S$ is the regularized standard deviation defined by $S(\boldsymbol{z}, \epsilon) = \sqrt{\text{Var}(\boldsymbol{z}) + \epsilon}$, $\gamma$ is a constant value set to 1 in our experiments, and $\epsilon$ is a small scalar that helps to prevent numerical instability. This term avoids dimensional collapse by maximizing the volume of the distribution of the mapped views in all dimensions. In other words, it prevents the well-known trivial solution where the representations of the two views of a sample collapse to the same representation (Jing et al., 2022).

**Covariance term**: The covariance regularization terms $C$ decorrelates different dimensions of the representations and prevents them from encoding similar information. The covariance matrix of $\boldsymbol{Z}$ is $C(\boldsymbol{Z}) = \frac{1}{n} \sum_{i=1}^{n} (\boldsymbol{z}_i - \bar{\boldsymbol{z}})(\boldsymbol{z}_i - \bar{\boldsymbol{z}})^T$ where $\bar{\boldsymbol{z}} = \frac{1}{N} \sum_{n}^{i=1} \boldsymbol{z}_i$. The covariance regularization term $c$ is then defined as the sum of the squared off-diagonal coefficients of the covariance matrix as follows $c(\boldsymbol{Z}) = \frac{1}{d} \sum_{i \neq j} [C(\boldsymbol{Z})]_{i,j}^2$ where $[C(\boldsymbol{Z})]_{i,j}^2$ is the element at row $i$ and column $j$ of the matrix $C(\boldsymbol{Z})$. Both the variance and covariance terms helps to maximize the information encoded by the model in the representation space.

**Invariance criterion**: The invariance criterion $s$ between $\boldsymbol{Z}'$ and $\boldsymbol{Z}''$ is defined as the mean squared Euclidean distance between the representation vectors in the two views $s(\boldsymbol{Z}', \boldsymbol{Z}'') = \frac{1}{n} \sum_i \|\boldsymbol{z}'_i - \boldsymbol{z}''_i\|_2^2$. The invariance term encourages the parametric mapping to ensure that the views of an object remain close in the latent space.

Finally, the SSL loss function $\mathcal{L}^{SSL}$ over a batch of representations is a weighted average of the invariance, variance, and covariance terms:

$$\mathcal{L}^{SSL} = l(\boldsymbol{Z}', \boldsymbol{Z}'') = \lambda s(\boldsymbol{Z}', \boldsymbol{Z}'') + \mu[v(\boldsymbol{Z}') + v(\boldsymbol{Z}'')] + \nu[c(\boldsymbol{Z}') + c(\boldsymbol{Z}'')] \tag{10}$$

In our experiments, we set $\lambda = \mu = 25$ and $\nu = 1$, following Bardes et al. (2022).

## A.2 Additional Results

### A.2.1 Effect of Temporal Edge Encodings on Baselines

Our proposed temporal edge encodings can theoretically improve the expressiveness of DyG2Vec (Souza et al., 2022). In Table 6, we investigate whether they can also improve the baselines' performance. That is, we add the temporal edge encodings outlined in Sec. 4.1 to some of the most powerful baselines, TGN (Rossi et al., 2020) and CaW (Wang et al., 2021b). Interestingly, we notice only minor improvements (1-2%) for some datasets and, in some cases, minor degradations. We hypothesize that this is due to the nature of baselines' encoder architectures. The use of RNNs or random walks to encoder history can make it difficult to effectively leverage the relevant temporal edge encodings across long time horizons.

Table 6: Effect of temporal edge encodings on baselines. All experiments are performed for one trial only in the FLP transductive setting. Temporal edge encodings are simply added as edge features to the baselines.

| Temporal Edge Encoding | Baseline | UCI | Enron | MOOC |
|---|---|---|---|---|
| × | TGN | **0.894** | 0.867 | 0.911 |
| ✓ | TGN | 0.879 | **0.872** | **0.936** |
| × | CaW | 0.930 | **0.970** | 0.937 |
| ✓ | CaW | **0.944** | 0.967 | **0.946** |

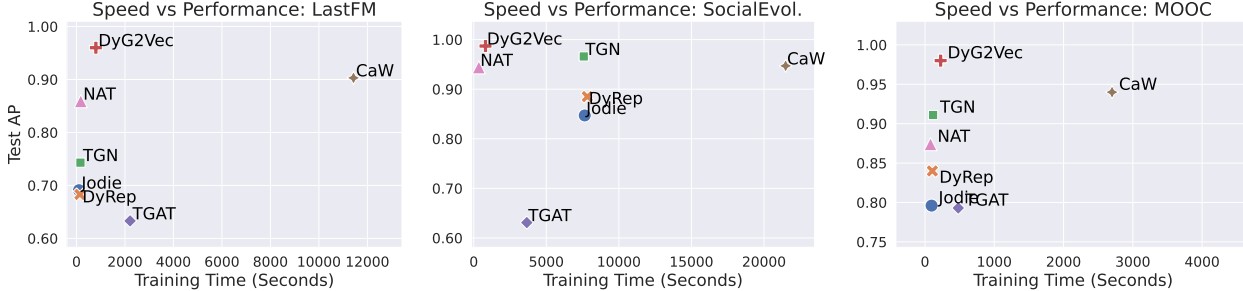

Figure 6: Transductive FLP performance (Test AP) vs Training Time (s) on 3 datasets. Training time represents the time it takes to train on the whole training set.

### A.2.2 Runtime and Computational Complexity

The main runtime overhead lies in how each of the baselines processes the input graph to predict a target edge. CaW samples $M$ $L$-hop random walks for each target edge. This is followed by an expensive set-based anonymization scheme. To achieve good performance, CaW can require relatively long walks (e.g., for Enron, $L = 5$). On the other hand, memory-based methods and TGAT sample a different $L$-hop subgraph for each target edge. DyG2Vec samples similar to TGAT but does so within a constant window size $W$ and without enforcing temporal causality.

Thus, assuming we use sparse operations in Pytorch Geometric (Fey & Lenssen, 2019) for message-passing, the encoding computational complexities are: DyG2Vec $= O(LW)$; CaW $= O(LMN_s)$ and TGN and variants $= O(LN_s)$. Here, $N_s$ represents the maximum number of sampled nodes in an $L$-hop subgraph. We can see that the main difference is the factor $M$. The factor $N_s$ comes from the complexity of message passing at each hop (assuming sparse operations). Note that DyG2Vec is limited to $O(W)$ nodes so it does not have this factor.

Table 7: Downstream Freeze test AP Results (after pre-training). DDGCL pre-training and downstream training were run with default parameters described in the work.

| Model | MOOC | Enron | UCI | LastFM |
|---|---|---|---|---|
| DyG2Vec | **93.1** | **96.6** | **95.4** | **93.0** |
| DDGCL | 84.3 | 83.0 | 85.3 | 78.8 |

### A.2.3 Comparison with Recent Dynamic Graph Models

GraphMixer (Cong et al., 2023b) uses a conceptually simple MLP-based link classifier that summarizes structural and temporal information from the latest temporal links and achieves surprisingly decent performance. Table 8 compares DyG2Vec with GraphMixer on transductive future link prediction. All results were generated under the same evaluation protocol in CaW (Wang et al., 2021b), using the DyGlib library *.

---

*https://github.com/yule-BUAA/DyGLib

Table 8: AP performance on Transduction future link prediction (Mean ± Std).

| Model | Wikipedia | Reddit | MOOC | LastFM | Enron | UCI | SocialEvol. |
|---|---|---|---|---|---|---|---|
| GraphMixer | $0.974 \pm 0.001$ | $0.975 \pm 0.001$ | $0.835 \pm 0.001$ | $0.862 \pm 0.003$ | $0.824 \pm 0.001$ | $0.932 \pm 0.006$ | $0.935 \pm 3e-4$ |
| **DyG2Vec** | $\mathbf{0.992 \pm 0.001}$ | $\mathbf{0.991 \pm 0.002}$ | $\mathbf{0.938 \pm 0.010}$ | $\mathbf{0.979 \pm 0.006}$ | $\mathbf{0.987 \pm 0.004}$ | $\mathbf{0.976 \pm 0.002}$ | $\mathbf{0.978 \pm 0.010}$ |

### A.2.4 Comparison to other dynamic Graph SSL Methods

As mentioned in Section 2, DDGCL Tian et al. (2021) proposed a contrastive SSL method for dynamic graphs that learns rich representations by contrasting node embeddings across time. Though experiments show improved performance on the future link prediction and dynamic node classification task, we believe the approach comes with several shortcomings that limit it's advantages in real world graphs. First, it is built on the TGAT Xu et al. (2020) encoder which, as seen in Table 2, is a weak encoder; particularly, for large datasets such as LastFM. Second, experiments for the FLP task are limited to the Reddit and Wikipedia datasets which are relatively easy. Lastly, the authors do not experiment under the standard settings in graph SSL literature such as the freeze and semi-supervised settings. Table 7 shows the results for downstream future link prediction under the freeze setting. The results show up to 10% gap compared to DyG2Vec, particularly for datasets where the TGAT encoder under-performs (e.g. Enron, UCI).

### A.2.5 SSL for Future Link Prediction

Table 9: Evaluating the performance of SSL pre-training using linear probing. Results are reported in AP (Mean ± Std) on Transductive Future Link Prediction.

| Setting | UCI | Enron | MOOC | LastFM |
|---|---|---|---|---|
| Random-init | $0.865 \pm 0.004$ | $0.913 \pm 0.007$ | $0.863 \pm 0.001$ | $0.817 \pm 0.002$ |
| Supervised | $0.988 \pm 0.007$ | $0.991 \pm 0.001$ | $0.980 \pm 0.002$ | $0.960 \pm 1e\text{-}4$ |
| SSL-init | $0.954 \pm 0.002$ | $0.966 \pm 0.001$ | $0.931 \pm 0.001$ | $0.930 \pm 2e\text{-}4$ |

Table 9 reports the transductive AP results for DyG2Vec under 3 different settings. Namely, we compare a random frozen encoder (Random-init) and an SSL pre-trained encoder (SSL-init) with the supervised baseline. Note that all variants use the DyG2Vec architecture. The difference lies in how the encoder is initialized and whether it is frozen during downstream training. SSL-init is initialized with a pre-trained encoder that is frozen during downstream training while "Supervised" is initialized randomly and is fully trained (encoder+decoder) on the downstream task. The results reveal that our SSL pre-training learns informative node embeddings that are almost on par with the fully supervised baseline. This supports the capability of the non-contrastive methods to learn generic representations across unlabelled large-scale dynamic graphs, which is in line with the findings for other data modalities (Bardes et al., 2022). The Random-init baseline is surprisingly good, as observed by recent works (Thakoor et al., 2022), but is outperformed by the SSL pre-trained encoder.

### A.2.6 Window-based Pre-training

In Table 10, we show the importance of window-based pre-training to learn the fine-grained temporal motifs of dynamic graphs. The "Full-graph" SSL setting represents applying the SSL loss on the full dynamic graph at once for a total of 300 epochs. Note that this is similar to the pre-training strategy used on static graphs and is difficult to scale for large scale graphs that do not fit to memory. The window-based strategy outperforms the full-graph mode for most datasets, particularly for large graphs (e.g. MOOC and LastFM) where we observe up to a 10% gap.

### A.3 Implementation Details

We train our model using the Pytorch framework (Paszke et al., 2019). The dynamic graph data and GNN encoder architecture are implemented using Pytorch Geometric (Fey & Lenssen, 2019). The ReLU

Table 10: Effect of Window-based pre-training on Linear probing AP results on Transductive FLP.

| SSL Setting | UCI | Enron | MOOC | LastFM |
|---|---|---|---|---|
| Window-based | **0.956** | 0.965 | **0.931** | **0.930** |
| Full-graph | 0.954 | **0.966** | 0.912 | 0.838 |

activation function is used for all models. The code and datasets are publicly available at `https://github.com/huawei-noah/noah-research/tree/master/graph_atlas`.

**Window-based framework**: As mentioned in Section 4, during SSl pre-training, the full dynamic graph $G_{0,E}$ is divided into a set of intervals $I$ that is generated by dividing the entire time-span into $M = \lceil E/S \rceil - 1$ intervals with stride $S$ and interval length $W$:

$$I = \left\{ \left[ \max(0, jS - W), \min(jS, E) \right) \mid j \in \{1, 2, \ldots, M\} \right\}. \tag{11}$$

Here, $W$ defines the number of edges in an interval and $S$ defines the stride. Note that we include all intervals up to but not including $[E - W, E)$ so that the target interval contains at least one edge.

**Decoder Architecture**: Denote by $t^{max}$ the timestamp of the latest interaction, within the provided history, incident to node $u$. For future link prediction, to predict a target interaction $(u, v, t)$, our decoder maps the sum of the two node embeddings of $u$ and $v$ and a time embedding of $t - t^{max}$ to an edge probability. Following Xu et al. (2020), the FLP decoder is a 2-layer MLP.

For dynamic node classification, to predict the label of node $u$ for interaction $(u, v, t)$, the decoder maps the source node embedding and time embedding of $t - t^{max}$ to class probabilities. Following Xu et al. (2020), the DNC decoder is a 3-layer MLP with a dropout layer with $p = 0.1$.

The time embedding is calculated using a trainable Time2Vec module (Kazemi et al., 2019). The time embedding allows the decoder to be time-aware; hence, possibly output different predictions for the same nodes/edges at different timestamps.

For SSL pre-training, the predictor $p_\phi$ is a simple 2-layer MLP that maps node embeddings $\boldsymbol{H}$ to node representations $\boldsymbol{Z}$.

**Distortion Pipeline**: We use the common edge dropout and edge feature dropout distortions. Both distortions are applied with dropout probability $p_d = 0.3$ which we have found to work best in a validation experiment exploring the values $p_d \in \{0.1, 0.15, 0.2, 0.3\}$. The edge feature dropout is applied on the temporal edge encodings introduced in Section 4.1, i.e., $z_p(t_p)$ and $c_p(t_p)$.

**Hyper-parameters**: We use a constant learning rate of 0.0001 for all datasets and tasks. DyG2Vec is trained for 100 epochs for both downstream and SSL pre-training. The model from the last epoch of pre-training is used for downstream training. For downstream evaluation, we pick the model with the best validation AP performance. Overall, we found that DyG2Vec converges within $\sim 50$ epochs.

For downstream training, We use a constant window size of $64K$ for all datasets except for MOOC, SocialEvolve, and Enron where we found a smaller window size of $8K$ works best. The batch size is set to 200 target edges. However, the model could be sped up by increasing batch size at the cost of higher memory. During SSL pre-training, we use a constant window size of 32K with stride 200.

Following previous work (Rossi et al., 2020; Xu et al., 2020), all dynamic node classification training experiments are performed with L2-decay parameter $\lambda = 0.00001$ to alleviate over-fitting.

## A.4 Baselines

**Baselines:** Following prior work (Rossi et al., 2020; Xu et al., 2020), all baselines are trained with a constant learning rate of 0.0001 using the Adam optimizer (Kingma & Ba, 2015) on batch-size 200 for a total of 50 epochs. The early stopping strategy is used to stop training if validation AP does not improve for 5 epochs.

For JODIE (Kumar et al., 2019), DyRep (Trivedi et al., 2019), and TGN (Rossi et al., 2020), we use the general framework implemented by Rossi et al. (2020). The node memory dimension is set to 172. For the NAT baseline Luo & Li (2022), we utilize the results in the paper for the common datasets since the setup is the same. We generate results for the missing datasets with the default hyperparameters. See Tables 12 and 11 for details.

For TGAT, we use the default hyperparameters of 2 layer neighbor sampling with 20 neighbors sampled at each hop. For the CaW method, we tune the time decay parameter $\alpha \in S$ where $S =$, and length of the walks $m \in \{2, 3, 4, 5\}$ on the validation set. The number of heads for the walking-based attention is fixed to 8.

Table 11: Hyperparameters for CaW.

| Dataset | Time Decay $\alpha$ | Walk Length $m$ |
|---|---|---|
| Wikipedia | 4e-6 | 4 |
| Reddit | 1e-8 | 3 |
| MOOC | 1e-4 | 3 |
| LastFM | 1e-6 | 3 |
| UCI | 1e-5 | 2 |
| Enron | 1e-6 | 5 |
| SocialEvolution | 3e-5 | 3 |

## A.5 Computing Infrastructure

All experiments were done on a Ubuntu 20.4 server with 72 Intel(R) Xeon(R) Gold 6140 CPU @ 2.30GHz cores and a RAM of size 755 Gb. We use a NVIDIA Tesla V100-PCIE-32GB GPU.

## A.6 Additional Related Work

**Self-supervised learning for dynamic graphs:** Jiang et al. (2021) adapt a sub-graph contrastive learning method (Jiao et al., 2020) where a node representation is contrasted in both structure and time. That is, for each node in the graph, a GNN encoder is trained to contrast its real temporal subgraph to its fake temporal subgraph . This is done by constructing a positive sample, a structural negative sample and a temporal negative sample. The positive sample is a time-weighted subgraph representation. The margin triplet loss is proposed to maximize the mutual information with the positive sample while maximizing distance with the structural and temporal negative samples. Experiments on downstream link prediction task under the freeze setting show improvement over baselines. However, their approach comes with several shortcomings. First, initial node features are computed as one-hot encodings which makes the method not suitable for the inductive scenario (i.e. predicting on new nodes). Second, the use of contrastive learning method is known to result in high memory and computation due to negative sampling (Thakoor et al., 2022). This makes the method less desirable for large-scale graphs. Third, they do not include results on other downstream tasks (e.g. dynamic node classification). Lastly, they do not compare to the SoTA CaW method (Wang et al., 2021b).

Table 12: Hyperparameters for NAT baseline. We use the default settings outlined in the paper Luo & Li (2022) for all other hyperparameters.

| Param | MOOC | LastFM |
|---|---|---|
| $M_1$ | 32 | 32 |
| $M_2$ | 16 | 16 |
| F | 4 | 4 |
| Self Rep. Dim | 32 | 72 |

Tian et al. (2021) adapt the TGAT encoder with a self-supervised contrastive framework across time. That is, they propose an extension to the classic contrastive learning paradigm by contrasting two nearby temporal views of the same node using a time-dependent similarity metric. Moreover, a de-basied contrastive loss is utilized to correct the typical negative sampling bias in contrastive learning. Experiments on the fine-tune and mutli-task learning settings show that the simple TGAT encoder can be significantly improved on both future link prediction and dynamic node classification. Nonetheless, their approach comes with several shortcomings. First, it is built on the TGAT encoder which, as seen in Tables 2, is a weak encoder; particularly, for large datasets. Second, experiments for the FLP task are limited to the Reddit and Wikipedia datasets which are relatively easy. Lastly, the authors do not experiment under the standard settings in graph SSL literature such as the freeze and semi-supervised settings. Table 7 shows the results for downstream future link prediction under the freeze setting. The results show up to 10% gap compared to DyG2Vec, particularly for datasets where the TGAT encoder under-performs (e.g. Enron, UCI).

Cong et al. (2023a) propose the dynamic graph transformer (DGT) which is a transformer-based graph encoder for *discrete-time dynamic graphs*. DGT is composed of two-tower networks that embed the temporal evolution and topological information of the input graph. Moreover, a temporal-union graph structure is proposed to efficiently summarize the temporal evolution into one graph. DGT is trained to encode the temporal-union graph using two complementary self-supervised pretext tasks. Namely, temporal reconstruction and multi-view contrasting. The first aims to reconstruct a snapshot given the past and present similar to how language models are trained. On the other hand, the latter is trained via non-contrastive learning on two views with randomly masked nodes. All together, DGT outperforms SOTA discrete-time baselines on several datasets for link prediction tasks. While they operate in a different domain, an interesting direction for future work would be to adapt their pre-training strategy for continuous-time dynamic graphs.

**More encoders for temporal graphs**: Souza et al. (2022) is a very recent work that establishes a series of theoretical results on temporal graph encoders. Their analysis exposes several weakness of both memory-based methods (e.g. TGN) and walk-based methods (e.g. CaW). Given these insights, they propose PINT, a memory-based method that leverages injective message-passing and novel relative positional encodings. The relative positional encodings count how many temporal walks of a given length exist between two nodes. Experiments show significant improvement over SoTA baselines on the link prediction task. However, the high theoretical expressive power comes at the cost of requiring relational positional encodings which are expensive to calculate in an online fashion. An interesting direction for future research would be to evaluate the expressive power of DyG2Vec compared to baselines using their theoretical framework (e.g. temporal WL test).

Wang et al. (2021a) adapt the vanilla transformer architecture to dynamic graphs by designing a two-stream encoder that extracts temporal and structural information from the temporal neighborhoods associated with any two interaction nodes. Rather than treating link prediction as a binary classification task, the authors leverage a contrastive learning strategy that maximizes the mutual information between the representations of future interaction nodes. Experiments show improved performance on future link prediction due to the more robust contrastive training strategy. Nonetheless, the paper does not compare to the SoTA CaW method (Wang et al., 2021b). Moreover, experiments are limited to the future link prediction task.

NeurTWs Jin et al. (2022) adapt causal walk sampling procedure of CaW Wang et al. (2021b) to include spatio-temporal bias and exploitation-exploration trade-off bias. Moreover, the authors propose utilizing ordinary differential equations (ODE) to better effectively model the irregularly sampled temporal interactions of a node and capture the underlying spatio-temporal dynamics. Experiments show good improvements over the baseline CaW model for some datasets. However, the addition of more sampling biases and ODEs comes at the cost of even higher computational complexity than CaW and more hyperparameters to tune, making the method undesirable for large real-world graphs.

TGL Zhou et al. (2022) introduce a general framework for training temporal GNNs on large-scale graphs. This is done by introducing an efficient CSR data structure to store temporal graphs and a parallel sampler that supports GPUs. The framework supports both memory-based methods (e.g. TGN Rossi et al. (2020)) and AMP architectures (e.g. TGAT Xu et al. (2020)). Experiments on billion-scale graphs show up to 13x

speed improvement while achieving the same performance. An interesting direction for future work is to adapt the DyG2Vec architecture into the TGL framework to further speed it up.

