# OpenReview forum: "DyG2Vec: Efficient Representation Learning for Dynamic Graphs"
_TMLR — Accepted by TMLR_

### Review · Reviewer_N3Q4 · 2023-10-04

**Summary Of Contributions:**

This manuscript studies the problem of dynamic graph SSL, where the dynamic graph is defined by the kind of graphs that may change their connections over time. The core of the proposed method includes temporal edge encodings and window-based subgraph sampling to generate the feature embedding, which is trained using the non-contrastive SSL similar to that in the computer vision field without a protector. Unfortunately, the writing's ambiguity makes it hard for me to make a summary of how the encoder is achieved.

**Audience:**

Yes

**Broader Impact Concerns:**

There are no Broader Impact Concerns for this manuscript.

**Claims And Evidence:**

No

**Requested Changes:**

My primary concern is the manuscript's writing quality, specifically its coherence and structure. I have detailed some of them in the "Strengths And Weaknesses" section, which the authors may reference.

**Strengths And Weaknesses:**

### Strengths

- It is indeed an interesting problem to learn the representation of the dynamic graph in a self-supervised manner.

- From a big picture of the proposed method, it seems to me promising to achieve self-supervised pre-training and downstream adaptation. However, given the unclear writing and organization of this manuscript, it is hard to check the detailed implementation.

-  The proposed window-wise construction for dynamic graphs makes sense to control the computational cost to a relatively constant value.

### Weakness

- The illustration and the capture of Fig.1 can be improved to make it more straightforward. Specifically,

    1)  the blue interval on the left side is too light to make it distinguished. There is also a dark blue block within the exact figure, making it hard to read at first glance.

    2)  symbols are not introduced well at the time it is first shown. The first reference to Figure 1 occurs in section 4.1 which is too far to be referred to by the readers. Similar problem for the FIgure. 2 and its reference. Reorganizing the narrative of the manuscript is highly recommended.

- For the problem formulation part, the meaning of symbols is improperly used. For example,

    1) The $E$ represents the number of edges but it is also used in $D^E$ to represent the dimension of the edge's feature. It is highly recommended to reformulate the equations and carefully adopt symbols.

    2) There is no symbol $z$ in Equation 1, do the authors mean $Z$?.

    3) $\theta$ represents the parameters of the encoder, but $\Theta$ is also the temporal edge encoder in Figure 1. It is recommended to use dissimilar symbols to represent different modules. Similar problems occur across the whole manuscript.

- To make the manuscript self-contained, it is necessary to provide a brief introduction to the definition of dynamic graph and the downstream tasks at the very beginning, as the term "dynamic graph" may refer to very different meanings within different sub-fields.

- I’m a little confused about the setup for the evaluation of the SSL for future link prediction shown in Table 4. If I understand in the right way, the SSL-init encoder also works with a trained downstream decoder to perform the task, is that right? Does the “Supervised” method have the same architecture as DyG2Vec? It is a little strange as the performance of the SSL-init one is lower than the pure supervised one if the SSL-init is fine-tuned for the downstream task, as this is one of the purposes of SSL.

- The descript of Figure 1 in Section 4.1 is not self-consistent. There is no “Temporal Attention Embedding” shown in Figure 1 and the “time-encoding” in Figure 1 is not introduced in this section. This unclear writing makes it hard to get the details about how the proposed method works.

---

### Review · Reviewer_ncve · 2023-10-18

**Summary Of Contributions:**

The present manuscript proposes a framework (DyG2Vec) for learning on continuous-time dynamic graphs. In particular, the core innovations in this work are (1) a new efficient attention-based neural network architecture, and (2) a modification of a well-known self-supervised paradigm (VICReg Bardes et al., 2022) for dynamic graphs that can be used for general-purpose pretraining. Additionally, (3) the authors make several engineering improvements compared to prior work (sampling temporal neighbours uniformly at random, enhancing neighbour messages with additional features – “temporal edge encoding”, window-based training strategy) that allows them to improve the efficiency as well as the performance of their architecture in downstream tasks. The method is conceptually and experimentally compared to several architectures for dynamic graphs on a variety of (large-scale) datasets and test setups (transductive vs inductive, link prediction, node classification) showing state-of-the-art performance in the majority of the cases. The comparative evaluation is complemented with ablation studies of various implementation details, providing justification for the choices made.

**Audience:**

Yes

**Broader Impact Concerns:**

I do not foresee any major ethical implication of this work, but I would encourage the authors to include a Broader Impact Statement for completeness, given that learning on dynamic graphs may have wide applicability.

**Claims And Evidence:**

Yes

**Requested Changes:**

-	**Motivation**. I think it would be helpful for the paper if the authors explain early on (in their introduction), why self-supervised learning is particularly important on dynamic graphs.
-	**Clarity**. The authors should aim to clarify the points I raised in my weaknesses above, i.e. both w.r.t. to their contributions, but most importantly w.r.t. to the experimental section and the various evaluation setups (this would be crucial for my recommendation). Additionally, section 4.3. has some minor clarity issues:
  - The node-level predictor is unclear. I think the explanation in Appendix A.2. should be moved to the main text. Also in section 3 the predictor for the SSL setup is denoted with $d_{\psi}$ instead of $d_\gamma$ (as in the “Prediction” paragraph).
  - Paragraph “SSL objective”: $s, \mu, c$ are undefined. I think this part should have been explained more thoroughly for those readers not familiar with the VICReg method.
-	**Missing experiments**. Similarly (this is also important for my recommendation), I would encourage the authors to complement their experiments with the ones suggested in the "weaknesses" section of my review. I would prioritise clarifying if the comparison is fair and if not, amend the experiments accordingly. Then, consistently comparing with/without the temporal edge encodings, and if possible, adding PINT and GraphMixer to the baselines.

**Minor**

-	Introduction: “We propose an effective message-passing encoder that leverages temporal edge encoding to increase its expressive power.” --> This is not theoretically proven, so it might be better to reduce this claim.
-	Since the temporal neighbours are sampled, I understand that the node embeddings are stochastic. I was wondering if the authors have measured their variance or the variance of the downstream task performance.
-	Why is $\bar{t}_i$, Eq. (7), calculated only on the sampled temporal neighbourhood and not on the entire one?
-	How crucial are the transformation distributions (section 4.3, paragraph “Views”)? Could the authors perform an ablation study on that? Also, the $t$ symbol denoting the transformations might be confusing since it is also used to denote timestamps.

**Strengths And Weaknesses:**

### Strengths
- **Presentation and contextualisation with prior work**. The paper is generally well-written and easy to follow, while the method is well-presented and relatively easy to understand. I also consider quite positive the fact that the related work is extensively covered and contrasted against the proposed method. It is also quite helpful for the reader in order to navigate the temporal graph learning landscape.
- **Technical soundness, execution and engineering choices**. Both the proposed architecture and the pretraining and downstream task strategies are reasonable and technically sound. At the same time, the above steps are relatively simple (unlike many previous works on temporal graph learning) and easy to implement, while the engineering choices are innovative and well-motivated.
- **Evaluation**. The experiments are convincing overall, and each building block is ablated and its existence is sufficiently supported by experimental evidence. Moreover, the training/inference runtime results are very encouraging, showing significant improvement w.r.t. efficiency/scalability. I also found the “attention weight analysis” (section 6) quite insightful.

Overall, the authors propose an original, well-executed, simple and efficient framework for self-supervised and supervised learning on continuous-time dynamic graphs with strong empirical performance.


### Weaknesses
-	**Clarity in the evaluation section**. Some parts of the experiments remain unclear which casts doubt on the fairness of the comparisons. In particular,
   - In section 5.2., Tables 2 and 3, the authors compare DyG2Vec against several temporal graph learning baselines on the tasks of “future link prediction” and “dynamic node classification”. However, it was unclear to me if DyG2Vec involves a self-supervised pretraining stage here or if it is trained end-to-end. In the case of the former, I was wondering if the baselines were also pre-trained. I believe that for fairness of comparison, the training strategy should be consistent across all methods, in order to clearly understand the benefits of either the pre-training or the new architecture.
   - It is quite hard to draw any safe conclusion from the performance reported on the dynamic node classification task. I believe the authors should provide more intuition as to why their method has significantly better performance in the 3rd dataset, while in the other two datasets, it is outperformed by the baselines. Additionally, I had trouble understanding this since I was unsure if the same pre-training & downstream task training strategy is conducted across all competing methods (see above).
   - In section 5.1., paragraph “Training protocols and hyperparameters”, the authors mention three different evaluation setups. However, I was unable to find where the results of these setups are actually reported. What setup do the authors apply in Tables 2 and 3?
   - Section 5.2, paragraph “SSL for future link prediction”. Comparing Table 4 to Table 2, I understand that the “Supervised” setup corresponds to the overall method proposed by the authors. Does this refer to end-to-end training (without pretraining) or SSL pretraining + fine-tuning on the downstream task? In case the former holds, then where are the benefits of the SSL pretraining shown? In case the latter holds, then what does “SSL-init” setup stand for? Moreover, how do these setups relate to the training protocols mentioned in a previous paragraph (see my comment above)? Could the authors explain these?
-	**Clarity of contributions**. Following up on my previous questions, I would like to ask the authors what they consider as their core contributions. Is it the self-supervised pre-training or the architecture + engineering choices (see summary)? I believe this should be more clearly stated in the paper and the reader should be pointed to the corresponding experimental evidence that provides justifications to the claims.
-	**Some experiments are missing**.
   - Depending on the answers to my above questions, perhaps the paper should be amended with extra experiments. E.g. in case Tables 2 and 3 compare fully-supervised baselines to a pre-trained DyG2Vec, then the experiments should be also performed under a consistent setup. In case they are all fully supervised, then more extensive results on pre-trained DyG2Vec should be reported.
   - Some baselines could be added: As far as temporal graph learning is concerned, I was wondering if the authors have compared to aggregation mechanisms similar to those used in PINT, Souza et al., NeurIPS’22 or GraphMixer, Cong et al., ICLR’23. As far as self-supervised pretraining is concerned, have the authors experimented with BYOL, Grill et al, NeurIPS’20 akin to BGRL, Thakoor et al., ICLR’22 for static graphs, as they nicely mention in their related work section? A few other works are also mentioned in the related work section (Tian et al., CIKM’21, Jiang et al., KDD’23 – the former is also briefly discussed experimentally in the appendix), so I tend to believe that more emphasis should be given to them in the experimental section of the main paper, e.g. by using the architecture + engineering choices of DyG2Vec and comparing only different SSL pretraining strategies.
   - I have a suspicion that the temporal edge encoding module might theoretically improve expressivity (mainly the “common neighbours” part). Therefore, I was wondering how the baselines would behave had the temporal edge encodings been used in there as well. Similarly (but less so), for the window-based strategy.

---

> ### Author Response · Authors · 2023-10-25
> **Author Response 1**
>
> > In section 5.2., Tables 2 and 3, the authors compare DyG2Vec against several temporal graph learning baselines on the tasks of “future link prediction” and “dynamic node classification”. However, it was unclear to me if DyG2Vec involves a self-supervised pretraining stage here or if it is trained end-to-end. In the case of the former, I was wondering if the baselines were also pre-trained. I believe that for fairness of comparison, the training strategy should be consistent across all methods, in order to clearly understand the benefits of either the pre-training or the new architecture.
>
> We apologize for the confusion. In Table 2, all models, including DyG2Vec, are trained end-to-end with no pre-training. In Table 3, following prior works, all models are pre-trained on the future link prediction task; see last sentence of "Training Protocols and Hyperparameters:" paragraph of Section 5.1
>
> > It is quite hard to draw any safe conclusion from the performance reported on the dynamic node classification task. I believe the authors should provide more intuition as to why their method has significantly better performance in the 3rd dataset, while in the other two datasets, it is outperformed by the baselines. Additionally, I had trouble understanding this since I was unsure if the same pre-training & downstream task training strategy is conducted across all competing methods (see above).
>
> As noted in our previous reply, all models use the same training strategy. We agree that the performance of DyG2Vec and the baselines is not consistent across all datasets. The DNC task by nature is a anomaly detection problem (e.g. predicting whether a user is banned) that has been adapted to node classification. The high class imbalance makes the model excessively confident. Prior works alleviate this issue by pre-training the encoder on future link prediction and freezing it during downstream training but we find that the performance issue still persists across all models. An interesting future direction would to design dynamic graph models tailored for this task.
>
> > In section 5.1., paragraph “Training protocols and hyperparameters”, the authors mention three different evaluation setups. However, I was unable to find where the results of these setups are actually reported. What setup do the authors apply in Tables 2 and 3?
>
> We apologize for the confusion. We have added references to the relevant tables (see changes in red). Table 2 represents the pure end-to-end supervised setting. Table 3 is the linear evaluation setting where the models are pre-trained on the future link prediction task.
>
> > Section 5.2, paragraph “SSL for future link prediction”. Comparing Table 4 to Table 2, I understand that the “Supervised” setup corresponds to the overall method proposed by the authors. Does this refer to end-to-end training (without pretraining) or SSL pretraining + fine-tuning on the downstream task? In case the former holds, then where are the benefits of the SSL pretraining shown? In case the latter holds, then what does “SSL-init” setup stand for? Moreover, how do these setups relate to the training protocols mentioned in a previous paragraph (see my comment above)? Could the authors explain these?
>
> "Supervised" represents full end-to-training with no SSL. "SSL-init" is initialized with a SSL pre-trained encoder that is frozen during downstream training. See clarifications in blue in section 5.2 paragraph "SSL for Future Link Prediction".
>
> > Clarity of contributions. Following up on my previous questions, I would like to ask the authors what they consider as their core contributions. Is it the self-supervised pre-training or the architecture + engineering choices (see summary)? I believe this should be more clearly stated in the paper and the reader should be pointed to the corresponding experimental evidence that provides justifications to the claims.
>
> The core contribution is a generic efficient architecture and engineering choices (e.g. window-based framework), that could be utilized for any dynamic graph learning task, including pre-training and downstream tasks.
>
> > Depending on the answers to my above questions, perhaps the paper should be amended with extra experiments. E.g. in case Tables 2 and 3 compare fully-supervised baselines to a pre-trained DyG2Vec, then the experiments should be also performed under a consistent setup. In case they are all fully supervised, then more extensive results on pre-trained DyG2Vec should be reported.
>
> Please see first reply. All models in a given table are evaluated under the same setup. Therefore, no further experiments are needed.

---

> > ### Author Response · Authors · 2023-10-25
> > **Author Response 2**
> >
> > > Some baselines could be added: As far as temporal graph learning is concerned, I was wondering if the authors have compared to aggregation mechanisms similar to those used in PINT, Souza et al., NeurIPS’22 or GraphMixer, Cong et al., ICLR’23. As far as self-supervised pretraining is concerned, have the authors experimented with BYOL, Grill et al, NeurIPS’20 akin to BGRL, Thakoor et al., ICLR’22 for static graphs, as they nicely mention in their related work section? A few other works are also mentioned in the related work section (Tian et al., CIKM’21, Jiang et al., KDD’23 – the former is also briefly discussed experimentally in the appendix), so I tend to believe that more emphasis should be given to them in the experimental section of the main paper, e.g. by using the architecture + engineering choices of DyG2Vec and comparing only different SSL pretraining strategies.
> >
> > We thank the review for raising this these works. We include a comparison with GraphMixer in Appendix A.2.2. As for the SSL pre-training, we believe DyG2Vec is compatible with any of the suggested static SSL loss functions. We chose VICReg for its simplicity and effectiveness. Experimenting with SoTA SSL methods would be an interesting future direction.
> >
> > > I have a suspicion that the temporal edge encoding module might theoretically improve expressivity (mainly the “common neighbours” part). Therefore, I was wondering how the baselines would behave had the temporal edge encodings been used in there as well. Similarly (but less so), for the window-based strategy.
> >
> > Indeed, the edge encoding can increase expressivity. Our ablation studies in Fig. 5 suggest that adding edge encoding and choosing the optimal widow size gives 1-2% boost in performance. Therefore, we believe the performance gap is also due to the architecture itself (e.g. no memory modules, shared message passing for src and target node, etc).
> >
> > > I think it would be helpful for the paper if the authors explain early on (in their introduction), why self-supervised learning is particularly important on dynamic graphs.
> >
> > SSL is important in any data modality to leverage large-scale un-labeled data. This is especially the case for dynamic graphs where labels are often scare and heavily imbalanced.
> >
> > > The node-level predictor is unclear. I think the explanation in Appendix A.2. should be moved to the main text. Also in section 3 the predictor for the SSL setup is denoted with $d_{\psi}$ instead of $d_\gamma$ (as in the “Prediction” paragraph).
> >
> > We have fixed the typos in the main text (See changes in red). The node-level predictor is simply a MLP. Unfortunately, due to space limitations, we are unable to move Sec A.2 to the main text.
> >
> > > Paragraph “SSL objective”: $s, \mu, c$ are undefined. I think this part should have been explained more thoroughly for those readers not familiar with the VICReg method.
> >
> > Thanks for raising this. We have added more details in Appendix A.1.
> >
> > > Similarly (this is also important for my recommendation), I would encourage the authors to complement their experiments with the ones suggested in the "weaknesses" section of my review. I would prioritize clarifying if the comparison is fair and if not, amend the experiments accordingly. Then, consistently comparing with/without the temporal edge encodings, and if possible, adding PINT and GraphMixer to the baselines.
> >
> > Please see table above for comparison with GraphMixer (Cong et al., ICLR'23). Regarding fair comparison, please see first 2 replies.
> >
> > > Introduction: “We propose an effective message-passing encoder that leverages temporal edge encoding to increase its expressive power.” --> This is not theoretically proven, so it might be better to reduce this claim.
> >
> > We have toned down the claim in the introduction.
> >
> > > Since the temporal neighbours are sampled, I understand that the node embeddings are stochastic. I was wondering if the authors have measured their variance or the variance of the downstream task performance.
> >
> > Our results in table 2 show little variance when changing the seed. Generally, we have not seen large variance in the embeddings or the performance.
> >
> > > Why is \bar{t}_i, Eq. (7), calculated only on the sampled temporal neighborhood and not on the entire one?
> >
> > We want to model to observe the temporal difference between the sampled edges only. Nonetheless, we believe either approach can work as long as the model can tell the temporal order of the edges.
> >
> > > How crucial are the transformation distributions (section 4.3, paragraph “Views”)? Could the authors perform an ablation study on that? Also, the � symbol denoting the transformations might be confusing since it is also used to denote timestamps.
> >
> > We use a constant dropout probability for all distortions (See Appendix A.3). We have not found the model to be over-sensitive to this parameter. The $t$ symbol has been changed to $d$.

---

> > > ### Comment · Reviewer_ncve · 2023-10-26
> > > **Post-rebuttal comments**
> > >
> > > I thank the authors for their response. In light of the clarifications, I would like to add some follow-up comments.
> > >
> > > - From the rebuttal clarifications, it is now relatively clear that *the main contribution is the architecture (perhaps mainly the temporal edge encoding) and the training tricks (window-based training strategy), rather than the self-supervised setup, with the bulk of the experiments (Tables 2 and 3) being conducted without self-supervision*. Therefore, I believe that the title and several parts throughout the paper are misleading, since, currently, one of the dominant messages conveyed to the reader is that this paper innovates on self-supervised learning for dynamic graphs.
> > > - Furthermore, *the self-supervised results are not convincing*. In particular, in Table 4, the "SSL-init" method (self-supervised pre-training --> freeze encoder --> decoder training on downstream task) performs worse than the "Supervised" method (end-to-end training), not to mention that the "Random-init" method (random frozen encoder --> decoder training on downstream task) is already quite competitive. This casts doubt on the effectiveness of the proposed self-supervised method, as one would expect the "SSL-init" method to be at least as performant as the "Supervised" one, to justify the extra computation required for pre-training.
> > > - *Perhaps the evaluation of the pre-training should be modified*. In Table 4, if I understand correctly, the input data used for pre-training and the downstream task are the same. Therefore, even if the representations learned by the self-supervised algorithm are indeed rich and discriminative, the chosen setup wouldn’t allow us to observe this (since there is no new knowledge to learn from and by freezing the encoder, we restrict the expressive power of the architecture, which in turn might result in worse downstream performance). In addition, Figure 4 is also a bit confusing. Do the authors use the same portion of the training data both for pre-training and fine-tuning? I believe it is more reasonable to pre-train on a different portion of the training data, so as to observe the aforementioned phenomenon (the discriminative power of the representations learned by pre-training). Also, it is necessary to compare with the end-to-end training method here, since it would be a better testbed to illustrate the benefits of the pre-training. Note that the motivation for self-supervised pre-training given by the authors in their response (“SSL is important in any data modality to leverage large-scale un-labelled data. This is especially the case for dynamic graphs where labels are often scare and heavily imbalanced.”), with which I agree, is not tested experimentally.
> > > - Overall, I am sceptical about the claims regarding the self-supervised pre-training stage  (“generic architecture […] including pre-training […] tasks”), since I could not find convincing empirical evidence that pre-training is actually beneficial. I think that either the self-supervised pre-training claims should be relaxed/removed, or should be carefully tested (including comparisons against other self-supervised methods).
> > > - The comparison against GraphMixer is indeed convincing. I encourage the authors to *also compare against PINT since it is one of the few works backed by sufficient theoretical evidence*. I would also like to reiterate that I believe *a comparison against baselines enriched with temporal edge encodings would make the experimental section more complete*.
> > > - *Minor*. I will agree with reviewer RAaR that the term “linear probing” is confusing. Personally, I believe the same holds for the term “semi-supervised probing”.

---

> > > > ### Author Response · Authors · 2023-10-26
> > > > **Author Response 3**
> > > >
> > > > We thank the review for the prompt response. Please find our response below.
> > > >
> > > > > From the rebuttal clarifications, it is now relatively clear that the main contribution is the architecture (perhaps mainly the temporal edge encoding) and the training tricks (window-based training strategy), rather than the self-supervised setup, with the bulk of the experiments (Tables 2 and 3) being conducted without self-supervision. Therefore, I believe that the title and several parts throughout the paper are misleading, since, currently, one of the dominant messages conveyed to the reader is that this paper innovates on self-supervised learning for dynamic graphs.
> > > >
> > > > Thanks for raising this up. As noted throughout the paper, the starting point of the main idea in this work is to show how a non-contrastive method can be adapted to dynamic graphs. The major contribution of the work is intervened between improving the representation learning of the dynamic graph encoder to improve the expressiveness power of the model while adapting it to project the input node features to the target output embeddings for any downstream task formulation including SSL pre-training. The novelty lies in the boundary of the two aspects while tilted more towards the encoding side from the formulation point of view. Findings in this work indeed reveal that the non-contrastive SSL methods that rely on volume maximization can help learn representations in an unsupervised manner while showing generalization on the downstream tasks under the two probing scenarios. The paper innovates on how non-contrastive methods can be adapted to dynamic graphs (i.e. what are the relevant data augmentations, does the SSL loss need to applied at node or sub-graph level, etc). Moreover, we investigate if VICReg is applicable and how much it is. The SoTA dynamic graphs models do not address how they can return node embeddings that are applicable to the SSL setup. Even if they do, investigation is needed to reveal if they are applicable to such pre-training setups. The majority of them rely on chronological training or require memory banks which brings challenges that must be addressed. Moreover, we improve the model by relying on window-based training and inference paradigm. This acts as a regularizer on the loss objective which modulates the model to learn to adapt to a proper level of interaction dynamics depending on the domain. The attention mechanism is formulated to encode the temporal interactions.
> > > >
> > > > > Furthermore, the self-supervised results are not convincing. In particular, in Table 4, the "SSL-init" method (self-supervised pre-training --> freeze encoder --> decoder training on downstream task) performs worse than the "Supervised" method (end-to-end training), not to mention that the "Random-init" method (random frozen encoder --> decoder training on downstream task) is already quite competitive. This casts doubt on the effectiveness of the proposed self-supervised method, as one would expect the "SSL-init" method to be at least as performant as the "Supervised" one, to justify the extra computation required for pre-training.
> > > >
> > > > Indeed, Random-init is a strong baseline. This is also the case in static graphs (See Table 3 in BGRL: "Large-scale Representation Learning on Graphs via Bootstrapping"). Static graph SSL methods have recently started to beat the supervised baseline after various research works attempted to improve the pre-training performance on the downstream tasks. Exploring further the graph SSL publications in recent years reveals that graph SSL is indeed at a very early stage and more attention from the community is needed to become as mature as the other modalities such as vision. Consequently, SSL on dynamic graphs is undergoing a worse situation. The purpose of this work is to pave the path ahead to show that we can gain insights on how the current dynamic graphs work under a non-contrastive SSL paradigm. Definitely, SSL pre-training in Computer Vision has been matured after a long track of research where Vision Transformers with higher model expressiveness have emerged and extremely heavy tuning of the entire pipeline on extremely large-scale datasets started to reveal that SSL pretraining evaluated under linear probing can supersede supervised training on the full dataset.

---

> > > > > ### Author Response · Authors · 2023-10-26
> > > > > **Author Response 4**
> > > > >
> > > > > > Perhaps the evaluation of the pre-training should be modified. In Table 4, if I understand correctly, the input data used for pre-training and the downstream task are the same. Therefore, even if the representations learned by the self-supervised algorithm are indeed rich and discriminative, the chosen setup wouldn’t allow us to observe this (since there is no new knowledge to learn from and by freezing the encoder, we restrict the expressive power of the architecture, which in turn might result in worse downstream performance). In addition, Figure 4 is also a bit confusing. Do the authors use the same portion of the training data both for pre-training and fine-tuning?...
> > > > >
> > > > > In a nutshell, we agree with the argument that the reviewer makes here. But the SSL community in other fields such as Computer Vision has adapted a similar setup where they pre-train on the entire dataset and probe on the same training set using the labelled data. For Figure 4, pre-training is done on the entire training data without accessing the target labels. Finetuning is done on a random portion of the labelled training data. This is to mimic the scenario where we have a lot of training data but few labels (a similar setup is used in the image domain where ImageNet is used for both the pre-training and downstream evaluations, see tables 6 and 7 in "A Simple Framework for Contrastive Learning of Visual Representations"). In dynamic graphs, the semi-supervised evaluation setup is very realistic in the context where a long history of interactions is maintained and only a small portion is required to be labelled. Here, SSL pre-training will reduce the cost of data annotation while maintaining the higher performance compared to random initialization and training on a small portion of annotated data
> > > > >
> > > > > > Overall, I am skeptical about the claims regarding the self-supervised pre-training stage (“generic architecture […] including pre-training […] tasks”), since I could not find convincing empirical evidence that pre-training is actually beneficial. I think that either the self-supervised pre-training claims should be relaxed/removed, or should be carefully tested (including comparisons against other self-supervised methods)
> > > > >
> > > > > We attempted to remain fidel to the SSL experimental setup on other data modalities to show how SSL pre-training is beneficial for dynamic graphs. The empirical results on the two probing scenarios are insightful to show that pre-training on dynamic graphs using non-contrastive methods can indeed improve the performance compared to the baseline methods. We agree that the comparison to other SoTA SSL methods is very insightful but we believe this is beyond the scope of this paper and could be a valuable future research direction to explore.
> > > > >
> > > > > > The comparison against GraphMixer is indeed convincing. I encourage the authors to also compare against PINT since it is one of the few works backed by sufficient theoretical evidence. I would also like to reiterate that I believe a comparison against baselines enriched with temporal edge encodings would make the experimental section more complete
> > > > >
> > > > > We include the PINT results (adopted from "Provably Expressive Temporal Networks") below.
> > > > >
> > > > > |            | Wikipedia      | Reddit        | MOOC           | LastFM         | Enron          | UCI            | SocialEvol.  |
> > > > > |------------|----------------|---------------|----------------|----------------|----------------|----------------|--------------|
> > > > > | GraphMixer | 0.974 ± 0.001  | 0.975 ± 0.001 | 0.835 ± 0.001  | 0.862 ± 0.003  | 0.824 ± 0.001  | 0.932 ± 0.006  | 0.935 ± 3e-4 |
> > > > > | PINT       | 0.987 ± .001   | 0.990 ± .001  | -              | 0.880 ± 0.007  | 0.887 ± 0.013  | 0.960 ± .001   | -            |
> > > > > | DyG2Vec    | 0.995 ± 0.003  | 0.996 ± 2e-4  | 0.980 ± 0.002  | 0.960 ± 1e-4   | 0.991 ± 0.001  | 0.988 ± 0.007  | 0.987 ± 2e-4 |
> > > > >
> > > > > > Minor. I will agree with reviewer RAaR that the term “linear probing” is confusing. Personally, I believe the same holds for the term “semi-supervised probing”
> > > > >
> > > > > We agree that the term "linear" is slightly misused. The reason we call it "linear" is to remain fidel to the literature and help the reader to quickly catch up on the details of the setup. We are going to replace all the usage of "linear probing" to "freeze probing" in the camera ready version to reflect the nature of the setup. Regarding semi-supervised probing, we refer the reviewer to the previous discussion we have on the details of the experimental setup for the semi-supervised probing. It reflects how practical this setup is when an endless stream of dynamic interactions are generated every second on the internet and it is not feasible to annotate the data at the same pace. We believe that the community can utilize SSL pre-training to improve the learned node representations such that only lightweight fine-tuning of the downstream task head is needed on limited labelled data to gain high generalization performance for the downstream task.

---

### Review · Reviewer_RAaR · 2023-10-19

**Summary Of Contributions:**

In this work the authors propose an efficient attention-based encoder architecture for temporal graphs - i.e. graphs where edges have an associated timestamp. Their proposed architecture also incorporates temporal edge encodings (which incorporate aspects such as node degree and number of common neighbors at a point in time) and a window-based subgraph sampling procedure (where, when making a prediction for a target edge $(u, v, t)$, the model will create a subgraph centered around $u$ and $v$ which contains edges in the time interval $(t-W, t)$.

The authors also propose a self-supervised pretraining objective for their encoder, which, given a sampled subgraph in a particular time window, creates two "distortions" (by applying edge dropout and feature masking), maps these distorted versions through the encoder to generate node embeddings, which are then put through a decoder which is trained to output node representations. The loss function is a regularization objective with 3 terms: an invariance term (to bring the two representations closer), a variance term (to avoid representation collapse), and a covariance term (to maximize information content).

They demonstrate that this self-supervised encoder training can result in improved results on future link prediction and dynamic node classification tasks. In these settings, new decoder architectures (MLPs) are fine-tuned for the particular task at hand. The authors show improved results on 7/7 datasets, while also being more efficient than methods which rely on sampling random walks and scaling more readily to large graph sizes (as opposed to methods which operated on a single static graph).

**Audience:**

Yes

**Broader Impact Concerns:**

The work is mostly of a methodological nature with no particular concerns from a broader impact perspective.

**Claims And Evidence:**

Yes

**Requested Changes:**

**R1 (critical):** Could you make the explanation of temporal attention (section 4.1) clearer? Specifically, here were a few things I had questions on:
1. The neighborhood $\mathcal N(i)$ has also not been defined here - is it temporally aware? Is it taken from a particular time interval? Or are we simply thinking of the temporal graph $\mathcal G$ as a multi-graph, and taking a neighborhood in the classic sense?
2. When you write $\mathcal N(i) = \\{e_p, \ldots, e_k\\}$, this is the first time we have seen the indices $p$ and $k$, and it's unclear to me what they are meant to represent. I think what was intended was to say that we uniformly sample some $N$ 1-hop neighborhood interactions of node $i$, $\\{e_{k_1}, \ldots, e_{k_N}\\} \subseteq \mathcal N(i) \subseteq \mathcal E$. (The indices $p$ and $k$ continue through the paper, so if I am correct than these should perhaps be changed elsewhere also.)
3. A minor note, but $\phi_p(t_p)$ also seems to depend on $l$, however it doesn't have the superscript the rest of the layer-dependent variables do.
4. Is $\Theta_p(t_p)$ a scalar? If so, it would be helpful to mention this.

In general, this section is presented with many undefined terms which are later clarified, which (personally) I find rather challenging to read because it requires the reader to maintain a lot of state while reading. Beyond the specific points raised above, I would encourage the authors to take another pass at this section.

**R2 (critical):** The discussion under the self-supervised loss (equation (9)) does not mention the $\mu$ term at all. Also, while the motivation for the various terms were provided, it would be helpful to present the actual functions used, at least in the appendix.

**R3:** In my experience, creating an evaluation for graph-structured data is quite challenging, and often the standard approach of simply comparing a known positive edge vs. a permuted negative is quite easy to "game" - for example, by capturing some high-level statistics - without actually learning anything important about the graph. I realize the primary evaluations in the paper are fairly standard, and I am not requesting the authors completely revamp the evaluation for this task, however as we already see performance seemingly saturate in Table 2 and the strong performance of the random initialization baseline in Table 4, it is worth considering whether the evaluation is simply too simple. I would like to know if the authors believe the models have actually learned something significant about the graph, or whether it is more likely that they have exploited some high-level statistics / bias in the test set? As an example of the sort of bias I mentioned - a pattern I have seen perform unreasonably well on other graph evaluations is to make any node which has a child in the training data be a parent to all nodes, and all other nodes have no children. On such datasets, neural models need only learn to be "parent classifiers", and can perform unreasonably well despite having learned almost nothing about the graph structure. To be clear, I'm not saying this is exactly what is happening here, but the eval numbers make me suspect that the eval set is simply too easy to tell us much of anything.

**Minor Typos / Suggestions:**
1. Section 4.2: I believe the decoder in this section should be labeled $d_\gamma$, not $d_\psi$.
2. Page 9: "performance due to difficult" -> "performance due to the difficult"
2. Page 10: "benefit from more embedding layers" -> delete?
3. It is mentioned in section 6 and in the caption of Figure 4 that the plot in the right of Figure 4 is a comparison of attention weights with respect to relative timespan. I assume this means the *mean* of the attention weights for a given timespan, correct?
4. In the section "Neighbor Sampling and Temporal Edge Encoding", it is mentioned that "sampling a single neighbor for higher hops gives SoTA performance compared to sampling 20 neighbors per hop (see 1st and 5th rows in Table 5)", and some . When I read this, I interpreted this to mean that "sampling more higher hops is better", which (after ingesting the results further) I realize is the opposite of the intended meaning. I would suggest conveying this by putting an emphasis on the number of samples in the first hop, rather than the 1 sample from the "higher hops".
5. Is "linear" really the right terminology to use when considering the fact that the decoders are MLPs? My understanding was that "linear probing" was just that - linear.

**Strengths And Weaknesses:**

*Strengths:*
* Demonstration that self-supervised pretraining works well with temporal graphs is useful contribution with practical implications in real-world applications.
* The model seems highly scalable and efficient, and empirical results support this.
* The writing of the paper was fairly clear, with the overall ideas and motivations being conveyed well.

*Weaknesses:*
* The mathematical details could be clarified in places. Some aspects, such as the SSL objective, are not fully specified (even in the appendix, unless I missed it). More concerningly, the main content of temporal
attention embedding was hard enough to follow that I am not confident I could reproduce the architecture from the description in the paper. (Specific questions in the requested changes.)
* While the current experiments highlight the strengths, it is possible that the evaluation tasks themselves are simply too easy.

---

> ### Author Response · Authors · 2023-10-25
> **Author Response**
>
> > R1 (critical): Could you make the explanation of temporal attention (section 4.1) clearer? Specifically, here were a few things I had questions on......
>
> We apologize for the confusion. All the aforementioned issues have been resolved. See changed in red and blue in section 4. Please note that, for generality, we define the encoder model given any input dynamic graph $\mathcal{G}$; however, as explained in Section 4.2, the input graph is restricted to a certain window of history.
>
> > R2 (critical): The discussion under the self-supervised loss (equation (9)) does not mention the $\mu$ term at all. Also, while the motivation for the various terms were provided, it would be helpful to present the actual functions used, at least in the appendix.
>
> Thank you for raising this. We have added more details in the Appendix A.1.
>
> > In my experience, creating an evaluation for graph-structured data is quite challenging, and often the standard approach of simply comparing a known positive edge vs. a permuted negative is quite easy to "game" - for example, by capturing some high-level statistics - without actually learning anything important about the graph. I realize the primary evaluations in the paper are fairly standard, and I am not requesting the authors completely revamp the evaluation for this task, however as we already see performance seemingly saturate in Table 2 and the strong performance of the random initialization baseline in Table 4, it is worth considering whether the evaluation is simply too simple. I would like to know if the authors believe the models have actually learned something significant about the graph, or whether it is more likely that they have exploited some high-level statistics / bias in the test set? As an example of the sort of bias I mentioned - a pattern I have seen perform unreasonably well on other graph evaluations is to make any node which has a child in the training data be a parent to all nodes, and all other nodes have no children. On such datasets, neural models need only learn to be "parent classifiers", and can perform unreasonably well despite having learned almost nothing about the graph structure. To be clear, I'm not saying this is exactly what is happening here, but the eval numbers make me suspect that the eval set is simply too easy to tell us much of anything.
>
> We agree with this point. Recent work [1] has shown that the negative sampling strategy can have a significant effect on the performance. As the experimental setup gets more mature, benchmarking DyG2Vec and baselines on such setups would be interesting future work. We chose to avoid deviating from the common metrics and evaluation setups used in all prior works. Nonetheless, we believe our variety of datasets provide an informative performance measure of the models.
>
>
> > Minor Typos / Suggestions: 1 - 5
>
> Thank you for raising these issues. All typos/suggestions have been resolved. See changes in red.
>
> > Is "linear" really the right terminology to use when considering the fact that the decoders are MLPs? My understanding was that "linear probing" was just that - linear.
>
> While it is correct that the decoders are non-linear, we chose to stick to the terminology used in prior SSL works to avoid confusion.
>
>
> [1] Towards Better Evaluation for Dynamic Link Prediction. Poursafaei, Farimah and Huang, Shenyang and Pelrine, Kellin and and Rabbany, Reihaneh. Advances in Neural Information Processing Systems 35 (NeurIPS 2022) Datasets and Benchmarks Track.

---

> > ### Comment · Reviewer_RAaR · 2023-11-20
> > **Thank you**
> >
> > Thank you for addressing my questions.
> >
> > I have reviewed the revised manuscript, however I feel the changes made thus far are not sufficient to completely address my concerns regarding clarity. Most of the original questions I have remain after the changes you have made. In my opinion, a more thorough rewrite of this section is necessary to meet the requirements of clarity for publication.

---

### Decision · Action_Editor_wdDM · 2023-11-28

**Recommendation:** Accept with minor revision

**Comment:**

The decision among the reviewers is mixed, with one reviewer in particular complaining about the lack of clarity which makes it hard for the reader to implement the proposed method, thus hindering reproducibility. The other two reviewers are in favour of accepting this submission.

Based on the reviews and discussion, the consensus is that the contributions of the paper are worth publication. A "minor revision" is recommended for this paper due to several remaining concerns. I request that the final version of the manuscript considers the final reviewers' comments, particularly the items listed below:
- Based on the discussion with reviewer ncve, it turns out that most of the experiments concern a fully-supervised setup, while the self-supervised method did not work that well in practice. Therefore, the focus of the paper should be mainly placed on the architecture and the training tricks, and not that much on the self-supervised setup. Τhe authors should update the title and the text of the paper to reflect that.
- Provide some explanations on why DyG2Vec outperforms the baselines. Ideally, explanations would be accompanied with some empirical results.
- Improve the quality of the presentation and clarity throughout the paper.
- As promised in the manuscript, make the code publicly available.

The authors are also encouraged to address the reviewers' concerns about the experimental setup. Specifically, it is not clear whether the experimental comparison is fair since the baselines do not utilize temporal edge encodings. The authors could investigate whether such features lead also the baselines to performance improvements.

**Audience:**

The paper will be of interest to at least a few individuals working on dynamic graphs.

**Claims And Evidence:**

The paper proposes DyG2Vec, an encoder-decoder model for continuous-time dynamic
graphs. Most claims made in the paper are supported by clear evidence. However, there are a few claims, such as that the model learns rich temporal unsupervised representations, which are not backed up by convincing empirical results.